# Positive Effect of *Lactobacillus acidophilus* EG004 on Cognitive Ability of Healthy Mice by Fecal Microbiome Analysis Using Full-Length 16S-23S rRNA Metagenome Sequencing

Soomin Jeon,[a] Hyaekang Kim,[a] Jina Kim,[a] Donghyeok Seol,[a,b] Jinchul Jo,[a] Youngseok Choi,[a] Seoae Cho,[b] Heebal Kim[a,b,c]

[a]Department of Agricultural Biotechnology and Research Institute of Agriculture and Life Sciences, Seoul National University, Seoul, Republic of Korea
[b]eGnome, Inc., Seoul, Republic of Korea
[c]Interdisciplinary Program in Bioinformatics, Seoul National University, Seoul, Republic of Korea

Soomin Jeon and Hyaekang Kim contributed equally to this article. Author order was determined retroalphabetically.

**ABSTRACT** Evidence for the concept of the "gut-brain axis" (GBA) has risen. Many types of research demonstrated the mechanism of the GBA and the effect of probiotic intake. Although many studies have been reported, most were focused on neurodegenerative disease and, it is still not clear what type of bacterial strains have positive effects. We designed an experiment to discover a strain that positively affects brain function, which can be recognized through changes in cognitive processes using healthy mice. The experimental group consisted of a control group and three probiotic consumption groups, namely, *Lactobacillus acidophilus*, *Lacticaseibacillus paracasei*, and *Lacticaseibacillus rhamnosus*. Three experimental groups fed probiotics showed an improved cognitive ability by cognitive-behavioral tests, and the group fed on *L. acidophilus* showed the highest score. To provide an understanding of the altered microbial composition effect on the brain, we performed full 16S-23S rRNA sequencing using Nanopore, and operational taxonomic units (OTUs) were identified at species level. In the group fed on *L. acidophilus*, the intestinal bacterial ratio of *Firmicutes* and *Proteobacteria* phyla increased, and the bacterial proportions of 16 species were significantly different from those of the control group. We estimated that the positive results on the cognitive behavioral tests were due to the increased proportion of the *L. acidophilus* EG004 strain in the subjects' intestines since the strain can produce butyrate and therefore modulate neurotransmitters and neurotrophic factors. We expect that this strain expands the industrial field of *L. acidophilus* and helps understand the mechanism of the gut-brain axis.

**IMPORTANCE** Recently, the concept of the "gut-brain axis" has risen and suggested that microbes in the GI tract affect the brain by modulating signal molecules. Although many pieces of research were reported in a short period, a signaling mechanism and the effects of a specific bacterial strain are still unclear. Besides, since most of the research was focused on neurodegenerative disease, the study with a healthy animal model is still insufficient. In this study, we show using a healthy animal model that a bacterial strain (*Lactobacillus acidophilus* EG004) has a positive effect on mouse cognitive ability. We experimentally verified an improved cognitive ability by cognitive behavioral tests. We performed full 16S-23S rRNA sequencing using a Nanopore MinION instrument and provided the gut microbiome composition at the species level. This microbiome composition consisted of candidate microbial groups as a biomarker that shows positive effects on cognitive ability. Therefore, our study suggests a new perspective for probiotic strain use applicable for various industrialization processes.

**KEYWORDS** *Lactobacillus acidophilus*, gut microbiome, gut-brain axis, cognitive ability, Nanopore sequencing

Address correspondence to Heebal Kim, heebal@snu.ac.kr.

D.S., S.C., and H.K. were employed by company eGnome, Inc. The remaining authors declare that the research was conducted in the absence of any commercial or financial relationships that could be construed as a potential conflict of interest.

The human body is a complex community that is colonized by various bacteria. Among the bacterial communities in the human body, the gastrointestinal tract (GIT) has the most abundant and varied bacterial community (1). In 2006, research showed that obesity is associated with bacterial composition in the gut, and therefore, a study for the gut microbiome began in earnest (2). The gut microbiome is defined as the collective genomes of microorganisms that live in the gastrointestinal tract. Functions of the gut microbiome have been reported, such as nutrient metabolism and regulation of the immune system for the host (3). The microbial composition in the gut is altered by environmental factors like age, diet, stress, and lifestyle, and a change in the microbial composition can induce physical changes in the host (4). Recently, the gut microbiome's effects on the brain have been proven and the concept of the gut-brain axis has risen to the surface (5). The gut-brain axis is a complex system involving the enteric nervous system and central nervous system, including the brain and spinal cord, and it works with bidirectional communication between the central and the enteric nervous system (6). Although the brain is located apart from the gut, the gut microbiome can affect the brain by stimulating the enteric nervous system and vagus nerve. Thus, dysbiosis of the gut microbiome often causes brain diseases. Recent experimental results described that gut microbiome dysbiosis was observed in patients with autism, Alzheimer's disease, and Parkinson's disease (7–10). At the same time, studies on mechanisms to understand the gut-brain axis were conducted. First, it was suggested that the microbial-derived metabolites are the main components acting on the neural pathways of the gut-brain axis (11, 12). The most well-studied substances are short-chain fatty acids (SCFAs), such as acetate, propionate, and butyrate, which are produced in the process of decomposing nondigestible fibers and carbohydrates (13). It promotes indirect signaling to the brain by modulation and induction of neurotransmitter and neurotrophic factors like $\gamma$-aminobutyric acid (GABA) and brain-derived neurotrophic factor (BNDF). Second, a study suggested that the gut microbiome affects brain function by regulating metabolic pathways (14). Previous research reported that the level of tryptophan metabolites, including serotonin and indolepyruvate, was altered by the gut microbiome. These metabolites have roles in the functioning of the gut-brain axis, such as signaling and antioxidant activities. Third, the gut microbiome may affect the brain by modulating the immune system (15). Interferon (IFN), tumor necrosis factor (TNF), and interleukin are well-known immune factors. According to recent studies, the amount of the immune factors is regulated by the intestinal microflora. These immune factors affect brain function by stimulating and activating the hypothalamic-pituitary-adrenal axis. Finally, it was suggested that gut microbes influence the brain directly by altering the fatty acid composition of the brain (16). Several studies have reported on the correlation between intestinal microbes and the brain, but the specific mechanism of the gut-brain axis is still not clear.

Probiotics are defined as bacteria that have positive effects on the host body (17). Probiotics have been used widely as a health supplement since they have various beneficial functions to host health with high adhesion properties to the intestine and low side effects. Most probiotics include bacterial genera that are Gram positive, facultative anaerobic, and rod shaped. *Lacticaseibacillus rhamnosus* is one of the longest-studied probiotic species, and many strains, such as LGG and GR-1 belonging to this genus, are available commercially. It is well known that *L. rhamnosus* has healing effects on diarrhea, acute gastroenteritis, and atopic dermatitis (18–20). Recently, its neurobehavioral effects, such as anxiety and depression relief, have been reported (21). *Lacticaseibacillus paracasei* is one of the representative probiotic species, and it has been shown to be effective for treating ulcerative colitis and allergic rhinitis (22, 23). In a recent study, an effect on age-related cognitive decline and a stress relief effect were reported with several strains of this species (24). *Lactobacillus acidophilus* is another representative probiotic bacterium that lowers cholesterol levels and has beneficial health effects, such as antibacterial effects against harmful bacteria like *Streptococcus mutans* and *Salmonella enterica* serotype Typhimurium (25, 26).

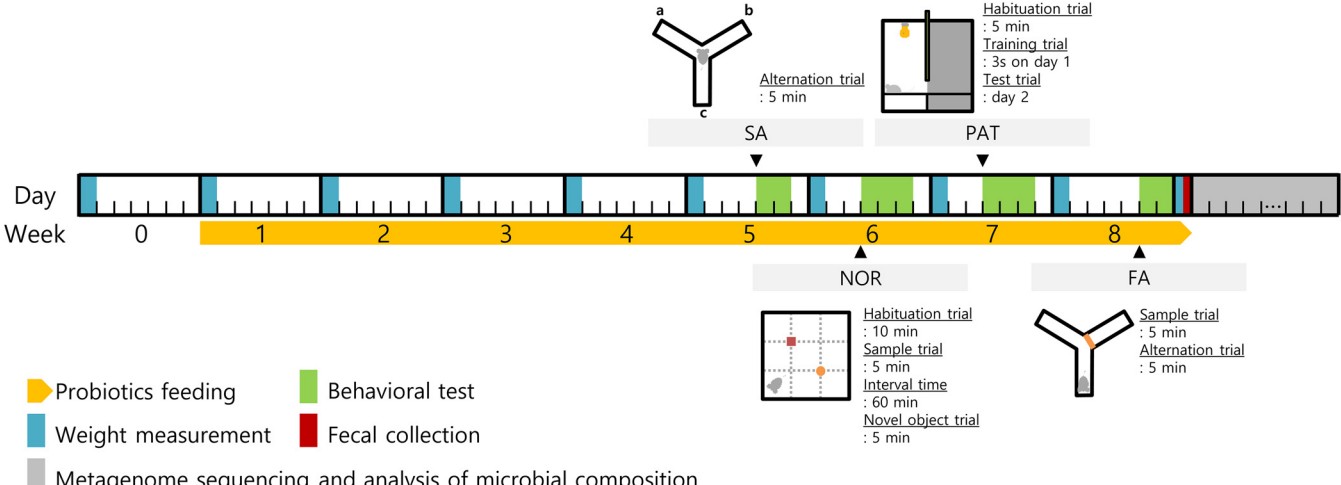

**FIG 1** Schematic diagram of the study conducted to discover a new probiotic strain that caused improved cognitive ability in mice. The diagram displays the experimental schedule by day and week for identifying a probiotic strain that improved cognitive ability. Cognitive ability was measured once a week by using four behavioral tests. The diagram of each experiment shows the first position of the animal.

In this study, we aimed to present a new strain that has an enhancing effect on cognitive ability through the gut-brain axis and provides an additional understanding of the gut-brain axis. Three probiotic bacteria, namely, *L. acidophilus*, *L. paracasei*, and *L. rhamnosus*, which have previously demonstrated beneficial effects to the host, were used to confirm their positive effects on host cognitive ability. Full 16S and 23S rRNA sequencing was performed to annotate the gut microbiome at the species level for downstream analysis. We expect our results to provide an understanding of the role of the gut microbiome.

## RESULTS

**Bacterial and animal treatments.** Three probiotic strains, namely, *L. acidophilus* EG004, *L. paracasei* EG005, and *L. rhamnosus* EG006, were identified by 16S rRNA sequencing. These strains were clustered with available *L. acidophilus*, *L. paracasei*, and *L. rhamnosus* strains, respectively, in a phylogenetic tree of the 16S rRNA gene (see Fig. S1 in the supplemental material). Probiotic strains were consumed by mice for 8 weeks with assessments of cognitive ability (Fig. 1). The averages of daily water intake per subject were similar between groups (Fig. 2A). Daily probiotic intakes were maintained constantly and the average amounts of the *L. acidophilus* group, *L. paracasei* group, and *L. rhamnosus* group were calculated as 7.82E09 ± 1.95E09, 4.37E10 ± 5.17E09, and 3.74E10 ± 3.98E09 CFUs (Fig. 2B). To identify the additional effect of probiotics, the

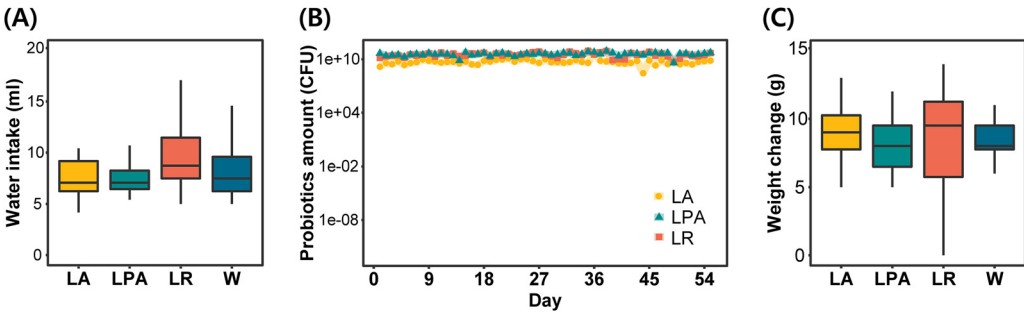

**FIG 2** Measurement of additional effects after probiotic consumption. Experimental groups are expressed in abbreviations. LA, *L. acidophilus* group; LPA, *L. paracasei* group; LR, *L. rhamnosus group*; and W, tap water-fed group (control). (A) The average daily water intake. All groups showed a similar average. (B) The change of daily probiotic amount by timeline. *L. acidophilus* was ingested in smaller amounts than the other two strains. (C) The average body weight change for 8 weeks. All groups showed similar averages.

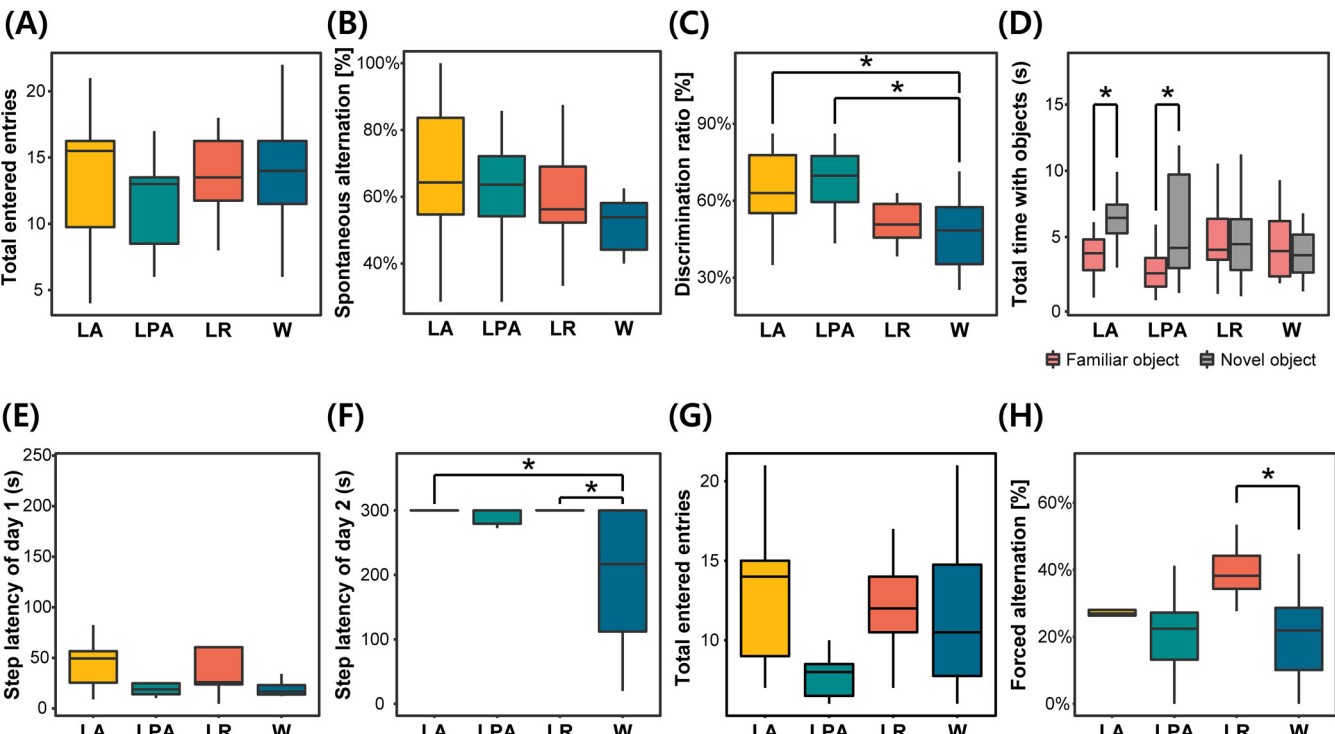

**FIG 3** Results of cognitive behavioral tests. Experimental groups are expressed in abbreviations. LA, *L. acidophilus* group; LPA, *L. paracasei* group; LR, *L. rhamnosus* group; and W, the group fed on tap water (control). (A) Total arm entries during spontaneous alternation test. (B) Spontaneous alternation. It is the representative value of the spontaneous alternation test. (C) Discrimination ratio. It is the representative value of the novel object recognition test. (D) Comparison of the total time to observe two objects. (E) Step-through latency of day 1. (F) Step-through latency of day 2. It is the representative result of the passive avoidance task. (G) Total arm entries during forced alternation test. (H) Forced alternation. This result is a representative value of forced alternation. All comparisons of averages between experimental groups were measured by the Wilcoxon signed-rank test. Significant difference is presented (*, adjusted *P* < 0.05).

body weights of mice were measured every week (Fig. 2C; see Fig. S2 in the supplemental material). Patterns of weight gain in the 4 groups were similar for 8 weeks. The mean body weight gains of the control group showed the highest value, which was 9.08 g. The *L. paracasei* group showed a significant difference from the control group with a *P* value under 0.05 in the second measurement, but the difference was recovered immediately. Similar to results of the weekly weight change, statistical significance was not found in accumulated weight between experimental groups for 8 weeks.

**Cognitive behavioral tests.** A spontaneous alternation test was conducted to assess spatial learning and short-term memory. Although the average number of the total entries to each arm in the *L. paracasei* group was slightly low, a difference between groups was not found (Fig. 3A). Mouse ratio showed spontaneous alternation for the first 3 entries; the *L. acidophilus* group showed the highest value as 75.0% (see Table S1 in the supplemental material). In spontaneous alternation, the average values of probiotic-fed groups were higher than the value of the tap water-fed group (Fig. 3B). Among the 4 experimental groups, the *L. acidophilus* group showed the highest alternation ratio. A Wilcoxon signed-rank test was performed to identify statistical significance, but there was no statistical difference between the experimental groups and control group.

A novel object recognition (NOR) test was performed to evaluate long-term and explicit memory using 4 different features (Fig. 3C and D; Table S1). The *L. acidophilus* group exhibited the highest average ratio of mice that touched the novel object before the familiar object, whereas the *L. rhamnosus* group showed the lowest value under the control group. For the discrimination ratio comparison, the three probiotic-fed groups showed higher average values than the control, and the *L. acidophilus* group

**TABLE 1** Metagenomic sequencing statistics of *L. acidophilus* group and control[a]

| Group | No. of samples | Total no. of reads | Estimated no. of bases (Mb) | $N_{50}$ (bp) | Total no. of counts | Total no. of OTUs |
|---|---|---|---|---|---|---|
| LA[b] | 5 | 312,384 ± 31,887 | 1,434 ± 143 | 4,872 ± 90 | 252,401.6 ± 25,171 | 528.4 ± 40 |
| W[c] | 5 | 335,356 ± 45,814 | 1,485.6 ± 215 | 4,748 ± 40 | 259,945.6 ± 35,117 | 539.8 ± 25 |
| Total | 10 | 323,870 ± 37,604 | 1,459.8 ± 173 | 4,810 ± 72 | 256,173.6 ± 28,860 | 534.1 ± 32 |

[a]There was no significant difference between groups. All values were presented as average ± standard error of the mean. Fecal samples compiled after 8 weeks of probiotic ingestion were used for metagenome sequencing.
[b]LA, *L. acidophilus* group.
[c]W, control group.

showed the highest values. To identify if there is a significant difference, a Wilcoxon signed-rank test was performed. Compared with the tap water-fed group, *L. acidophilus* and *L. paracasei* groups displayed statistically significant differences with an adjusted *P* value of 0.037. To identify animal behavior details, the number of objects touched and the total time of object observation in each group were compared. In a comparison of object touch, statistical differences were significant in *L. acidophilus* and *L. paracasei* groups with *P* values of 0.031 and 0.042, respectively. Also, the *L. acidophilus* group had a significant difference between the time taken to observe the familiar object and that of the novel object.

A passive avoidance task (PAT) was conducted to measure long-term and implicit memory. Step-through latency was used to compare the mean difference between the experimental groups. Most of the subjects were transferred into a darkroom for a minute on day 1 (Fig. 3E). Only 3 animals took more than 100 s to get into the darkroom. A difference between the experimental group and the control was not found on day 1. Compared with the latency time on day 1, the average latency time increased on day 2, and unexpectedly, 26 animals stayed in the lightroom for over 300 s (Fig. 3F). The *L. rhamnosus* group presented the highest average latency time, followed by the *L. acidophilus* group. while the control group showed the lowest average (Table S1). The Mann-Whitney U test was conducted to check the mean difference; the *P* values of *L. acidophilus* and *L. rhamnosus* groups were less than 0.05 compared with the control group. The adjusted *P* values of both groups were 0.040.

To assess spatial learning and long-term memory, forced alternation was conducted. Memory was evaluated by forced alternation (%), the number of arms that the mouse entered, and the percentage of mice in a group that entered the novel arm as their first entry. While the total number of the entries into each arm was diverse, there was no significant difference between the experimental groups and the control (Fig. 3G). The *L. acidophilus* group scored the highest ratio of mice entering the novel arm as their first entry (Table S1). Forced alternation values of *L. acidophilus* and *L. rhamnosus* groups were higher than the value of the control group (Fig. 3H). Forced alternation of the *L. rhamnosus* group and the control group had a significant difference with an adjusted *P* value of 0.038.

**Full 16S-23S rRNA sequencing and biological diversity.** Metagenome sequencing was performed with *L. acidophilus* and control groups, which showed the most improvement in cognitive ability. We compared the microbial composition of both groups. Gut microbial component information annotated at the species level was completely constructed by sequencing the entire 16S-23S rRNA of the mouse stool (Table 1). An average of 323,870.0 ± 84,085.5 reads were generated from 10 stool samples. The total number of identified operational taxonomic unit (OTUs) was 252,401.6 ± 56,284.7 in the *L. acidophilus* group and 259,945.6 ± 78,526.0 in the control group. The produced OTUs were annotated as a total of 528.4 ± 90.4 species in the *L. acidophilus* group and 539.8 ± 55.4 species in the control group. To check the sufficiency of the sequencing depth for the analysis, a rarefaction curve was created (Fig. 4A).

Alpha diversity was calculated to compare species richness within a group (Fig. 4B). In the comparison of the two groups, no significant difference was found in Chao1 and Shannon indexes. Beta diversity was measured to compare the diversity of the microbial

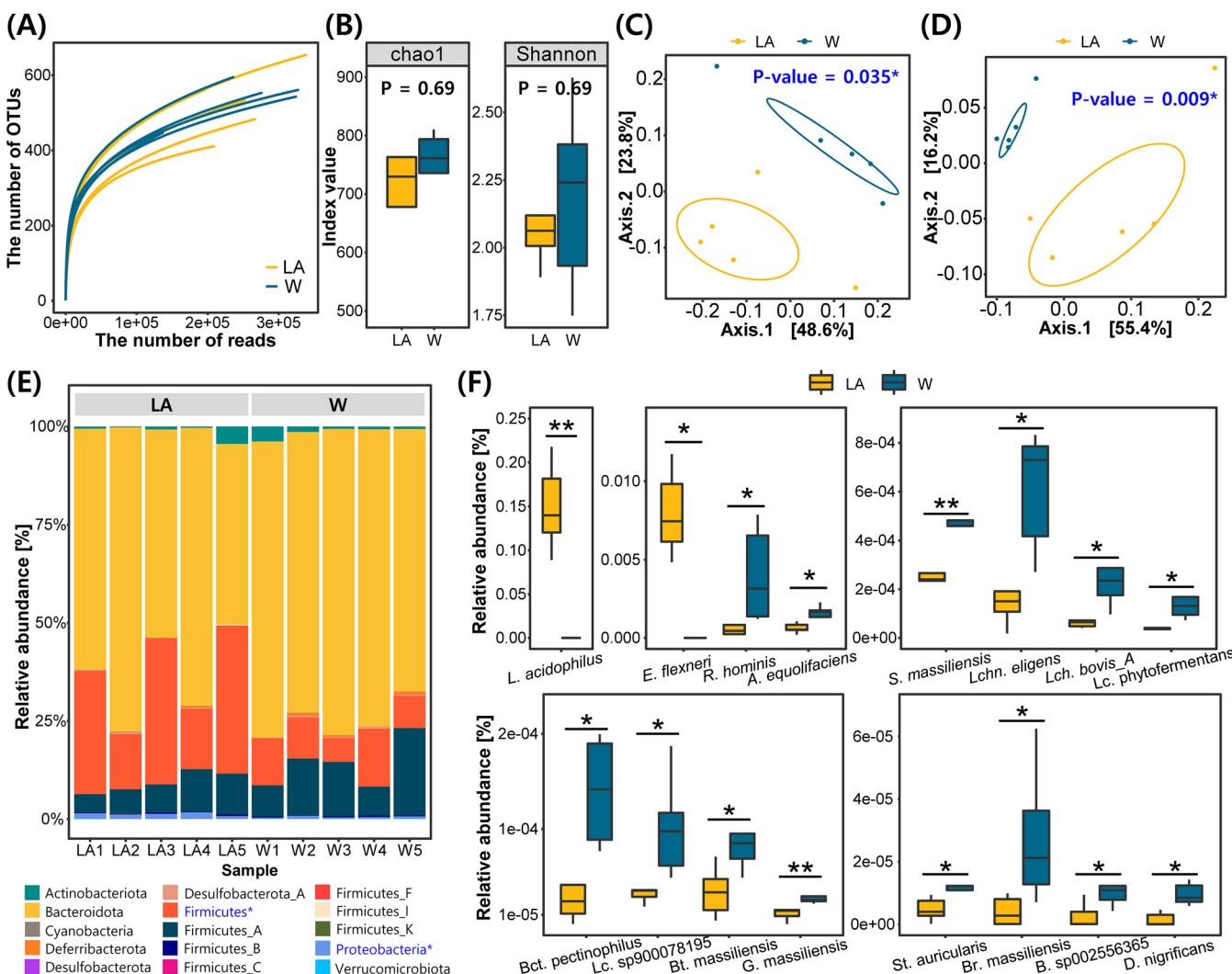

**FIG 4** Results of metagenomic sequencing. Experimental groups are expressed in abbreviations. LA, *L. acidophilus* group; and W, the group fed on tap water (control). (A) Rarefaction curve of metagenome sequencing. (B) Alpha diversity of the *L. acidophilus* group and control. (C) Beta diversity using Bray-Cutis distance between the *L. acidophilus* group and control. (D) Beta diversity using Unifrac distance between both groups. (E) Comparison of microbial composition at the phylum level. The blue-colored phylum with the * symbol showed a significant difference compared to the two experimental groups. (F) Comparison of microbial composition at the species level. All comparisons of the average between experimental groups were measured by independent *t* test. Significant differences are presented (*adjusted $P < 0.05$; **, $P < 0.01$).

community between the two groups (Fig. 4C and D). It was confirmed that both beta diversity evaluations (Bray-Cutis and Unifrac distance) had significant differences.

**Microbial composition.** In the comparative analysis of microbial compositions, taxonomies with significantly different ratios were found between the *L. acidophilus* group and the control group. At the phylum level, *Bacteroidota* accounted for the highest proportion of bacteria in both groups, followed by *Firmicutes* (Fig. 4E). Significant differences between the 2 groups were found in 2 of the 12 phyla (*Firmicutes* and *Proteobacteria*), of which all were high in the *L. acidophilus* group. At the class level, *Bacteroidia* showed the highest proportion of bacteria in both groups. Also, the proportion of *Bacilli* and *Gammaproteobacteria* classes were increased in the *L. acidophilus* group compared with those in the control group (see Fig. S3 in the supplemental material). At the order level, *Bacteroidales* showed the highest percentage of bacteria in both groups, and *Lactobacillales* and *Enterobacterales* orders were found to exhibit higher proportions in the *L. acidophilus* group. At the family level, *Muribaculaceae* showed the highest proportion of bacteria in both groups. It was found that 2 families (*Lactobacillaceae* and *Enterobacteriaceae*) showed increased proportions in the *L. acidophilus* group, while a

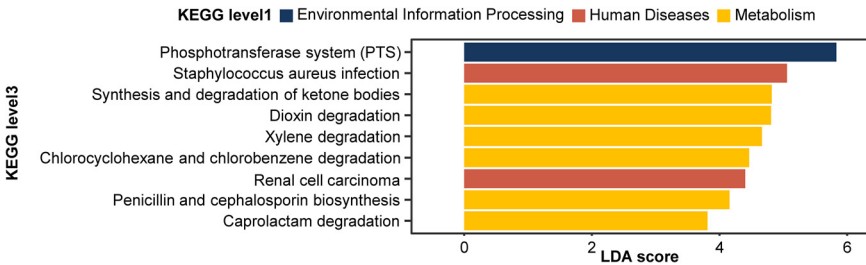

**FIG 5** Results of functional profiling. Predictive functional profiling of the microbiome. All predicted functions have a positive LDA score for the *L. acidophilus* group.

decreased percentage was observed in one family (*Ruminococcaceae*). In the genus comparison, *Muribaculum* showed the highest ratio in the 2 groups, and 12 genera showed differences between groups. Three genera showed an increased proportion in the experimental group, whereas 9 genera showed higher mean values in the control group. The genera increased in the *L. acidophilus* group were *Lactobacillus*, *Staphylococcus_A*, and *Escherichia*, whereas the genera decreased in the *L. acidophilus* group were *Bacteroides_F*, *Desulfotomaculum*, *Lachnobacterium*, *Bittarella*, *Agathobacter*, *Roseburia*, *Bariatricus*, and *Lachnospirarea*. At the species level, *Muribaculum intestinale* was found to account for the largest proportion of bacteria, with over 50% in both groups. Following *M. intestinale*, the species *Lactobacillus acidophilus*, *Lactobacillus johnsonii*, *Lactobacillus_B murinus*, and *Lactobacillus_H reuteri* were found with a high proportion in the *L. acidophilus* group, while *Lactobacillus_B murinus*, *Bacteroides_B vulgatus*, *Faecalibaculum rodentium*, and *Kineothrix alysoides* species showed a high proportion in the control group. No unique bacterial species were found in either of the two groups. Seventeen species showed differences between groups, and it was confirmed that the proportions of *L. acidophilus* and *Escherichia flexneri* were increased in the *L. acidophilus* group (Fig. 4F).

**Functional profiling and correlation analysis.** Functional profiling was performed at the KEGG level 3 to estimate the effect of the differential composition of intestinal microbes on the mice (Fig. 5). By calculating the linear discriminant analysis (LDA) score, we confirmed that the two groups showed significantly different patterns in 9 categories. All nine categories were predicted to be more activated in the *L. acidophilus* group. The phosphotransferase system (PTS) scored the highest followed by *Staphylococcus aureus* infection, synthesis, and degradation of ketone bodies.

To further estimate the influence of the altered gut microbiota, Spearman's correlation analysis of cognitive behavioral abilities and bacterial OTUs and fermentation products were performed (Fig. 6). *L. acidophilus* and *E. flexneri* showed a positive correlation with all assessments of cognitive abilities, while the other 14 OTUs presented a negative correlation. In particular, step-through latency at day 2 and step latency difference for 2 days of the PAT results showed a significant negative correlation with the *Gemella massiliensis* (r = $-0.8379$, $P = 0.03248$ and r = $-0.8182$, $P = 0.0376$) and *Desulfotomaculum nigrificans* (r = $-0.8781$, $P = 0.01914$ and r = $-0.8450$, $P = 0.03225$).

To provide evidence to indirectly infer the mechanism of action of the gut microbiome, the concentration of SCFA in the microbial culture was measured (see Table S2 in the supplemental material). Lactic acid and acetic acid were found in three microbial cultures. Lactic acid was identified in the highest concentration in *L. paracasei* EG005, and acetic acid was included in the highest concentration in the *L. acidophilus* EG004 culture. Propionate and butyrate were not within detectable ranges.

**Comparative analysis of genetic contents in bacterial whole-genome sequences.** To identify its safety and functionality, several genetic factors were detected. Fourteen genomic islands, 2 prophage regions, 1 CRISPR region, and 3 bacteriocins were found in the genome of *L. acidophilus* EG004. In *L. paracasei* EG005, 29 genomic islands, 7 prophage regions, 3 CRISPR regions, and 2 bacteriocins were detected (see Fig. S4 to S6 in

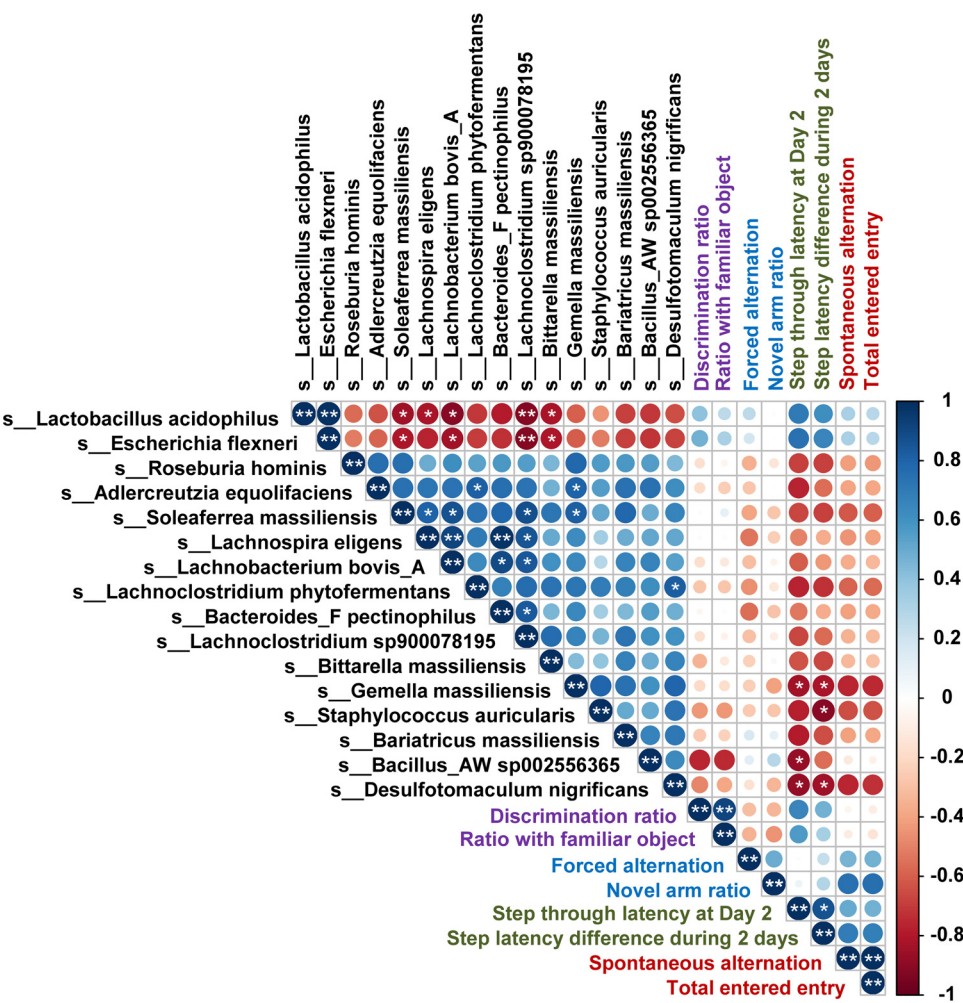

**FIG 6** Spearman's rank correlation analysis. A correlation analysis was conducted to detect an association among bacterial OTUs, measured cognitive abilities, and fermentation products. The color intensity and circle size show the strength of the correlation. Red color represents a negative correlation, and blue color is a positive correlation. Only circles with adjusted *P* value under 0.01 are illustrated in the matrix. The results of cognitive ability evaluation were classified by 4 colors: purple, NOR; blue, FA; deep green, PAT; and brown, SA. *, $P < 0.05$; **, $P < 0.01$.

the supplemental material). In the case of *L. rhamnosus* EG006, 23 genomic islands, 8 prophage regions, 3 CRISPR regions, and 1 bacteriocin were found in the genome. To estimate a genetic factor related to cognitive ability, protein annotation was conducted (Fig. 7A). Protein metabolism, carbohydrates, amino acids, and derivatives showed high proportions, but there was a difference in order by bacterial strains. Protein metabolism had the highest proportion in *L. acidophilus* EG004, and carbohydrates presented the highest proportion in *L. paracasei* EG005 and *L. rhamnosus* EG006. In a subcategory comparison of predicted functional sequences, a difference of genetic contents was found (Fig. 7B). Coding DNA sequences (CDSs) related to fatty acids were found in the genomes of *L. paracasei* EG005 and *L. rhamnosus* EG006. Genes of 3 subcategories (aromatic amino acids and derivatives; alanine, serine, and glycine; and proline and 4-hydroxyproline) were detected in *L. rhamnosus* EG006, while genes of 3 other categories in amino acids in derivatives were contained in only *L. acidophilus* EG004.

## DISCUSSION

As interest in the gut-brain axis has increased, many types of research in this

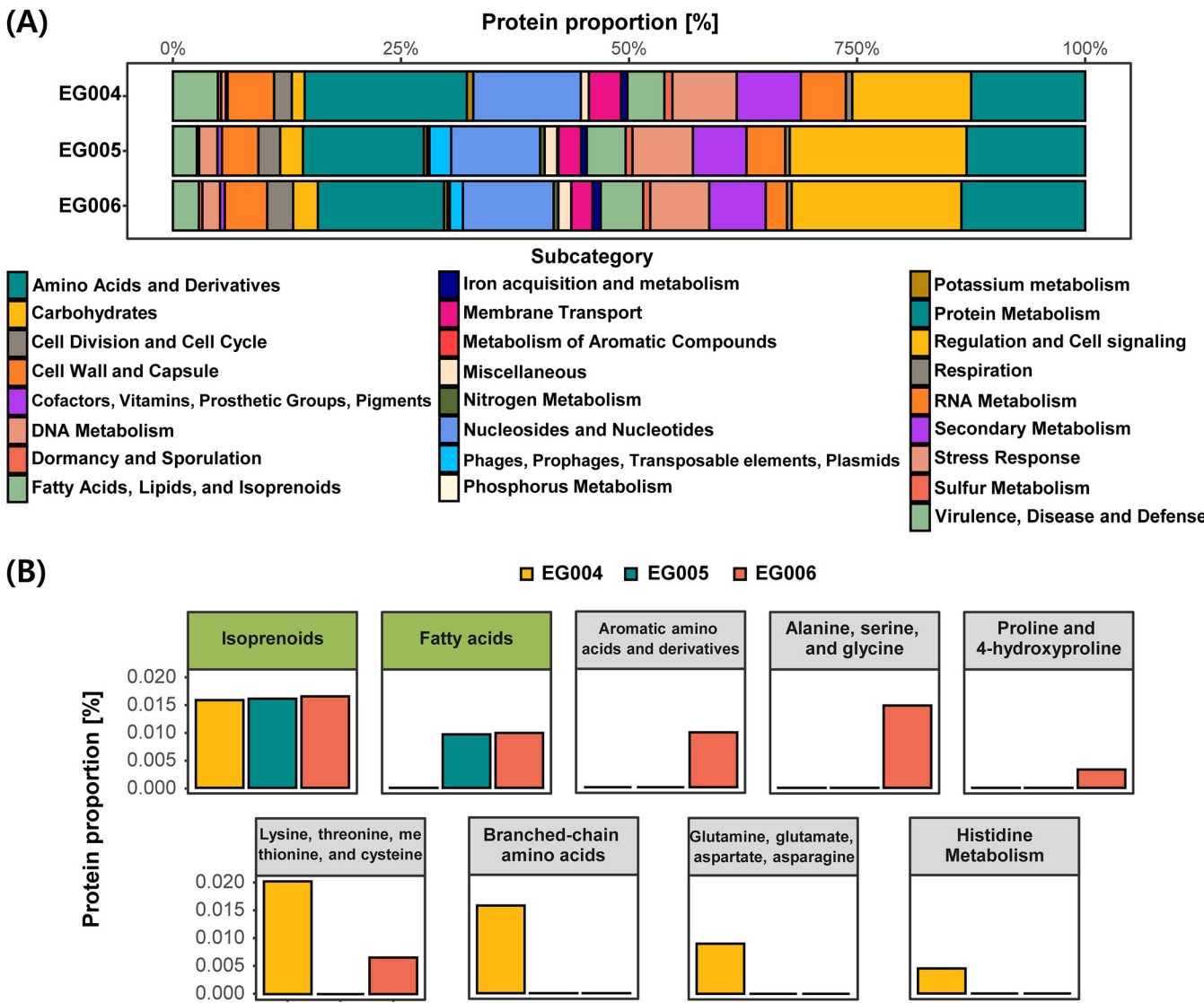

**FIG 7** Genomic comparison of 3 probiotic strains. (A) Functional classification of protein-coding sequences. All predicted protein sequences were classified by categories by the SEED system. (B) Subcategories in fatty acids, lipids, and isoprenolds and amino acids and derivatives. The fatty acids, lipids, and isoprenolds subcategory showed a yellow-green title box and amino acids and derivatives category presented a light-gray title box.

criterion have been published. However, it is still unclear about the integral mechanism and which strain has a positive or negative effect. Therefore, we aimed to develop a new strain that has a positive effect on host cognition, and we found 3 strains that caused positive effects in 4 different cognitive tests (Fig. 3). All experimental groups fed on the probiotic strains appeared to have improved cognitive ability. The group fed on *L. acidophilus* showed the highest score with a total score of 26, while the groups fed on *L. rhamnosus* and *L. rhamnosus* scored 16 and 15 points, respectively, which are slightly higher than that of the control group (see Table S3 in the supplemental material). The *L. acidophilus* group showed high scores in all evaluations, while others were highly evaluated only in some experiments. In addition, although probiotic consumptions were carried out as the same method, three experimental groups showed improved cognitive ability in different tests. This finding implies that different probiotic strains affect cognitive ability by different mechanisms and that *L. acidophilus* had an effect on a wider area than other strains. The *L. paracasei* group showed improved cognitive ability in the novel object recognition test. Previous studies indicated that this bacterium improves cognitive ability and increases the level of

serotonin and brain-derived neurotrophic factor (BDNF) in the hippocampus (24). Another bacterium, *L. rhamnosus*, displayed improved cognitive ability in the passive avoidance task and forced alternation test. Several studies demonstrated that *L. rhamnosus* consumption could increase cognitive ability by activating microglia in the hippocampus (27, 28). Similar to previous studies, we experimentally confirmed that *L. paracasei* and *L. rhamnosus* could enhance cognitive function. On the other hand, although it is indicated that the *L. acidophilus* strain has a neuroprotective effect against traumatic brain injury, there was no experimental research related to its cognitive ability (29, 30). In our study, we identified that the *L. acidophilus* group presented the highest classical measured values as well as incidental measured values in novel object recognition tests and passive avoidance tasks. This result indicates that *L. acidophilus* is capable of improving cognitive ability in a similar way as that of previously reported probiotics such as *Lcb. paracasei* and *Lcb. rhamnosus*. Our results will help further broaden the industrial field of probiotic strains.

To understand the effect of the gut microbiome on the brain as our secondary goal, we performed a gut microbiome analysis of *L. acidophilus* group, which showed the best cognitive improvement, along with the control group, The difference of species richness was not found in the comparison of alpha diversity, whereas the difference was found in the comparison of beta diversity (Fig. 4B and C and 4D). This finding suggests that the number of OTUs constituting the two gut microbial communities is similar but that the composition of the OTUs is different. In the comparison of the two communities, significant differences were observed at all taxonomic levels except for the bacterial kingdom. Naturally, the *L. acidophilus* group was confirmed to show a significant increase in *L. acidophilus* abundance and ultimately showed a high ratio of *L. acidophilus*. This result indicates that a large amount of *L. acidophilus* is capable of safely reaching the intestines without being affected by digestive juices, such as gastric acid and pancreatic enzymes.

We estimated that the positive effect on cognitive ability due to the increased proportion of *L. acidophilus* in the intestines was based on two rationales, as follows: modulation of neurotransmitters and neurotrophic factors and production of SCFAs. First, *L. acidophilus* modulates several types of neurotransmitters in the intestine. Microbial-derived intermediates, which affect the brain through gut epithelial and blood-brain barriers, are GABA (γ-aminobutyric acid), glutamate, dopamine, noradrenaline, serotonin (5-hydroxytryptamine [5-HT]), and brain-derived neurotrophic factor (BNDF). These neurotransmitters are synthesized from various amino acids. GABA and glutamate are produced from the gut microbiome, such as members of *Bifidobacterium* and *Lactobacillus* (31). Glutamate has a role as a neurotransmitter by itself, and it is used at GABA synthesis (32). Dopamine and noradrenaline are synthesized from specific amino acids, such as tyrosine and phenylalanine (33). L-Tryptophan is a well-known precursor of serotonin (34). Therefore, altered amino acid composition by the gut microbiome seems to affect the neurotransmitter synthesis in the host. In the comparison of the functional protein genes, *L. acidophilus* EG004 showed a higher composition of the gene related to amino acid metabolism than *L. paracasei* EG005 and *L. rhamnosus* EG006 (Fig. 7A). Changes in intestinal amino acid composition caused by ingested *L. acidophilus* may have led to differences in cognitive ability. It has been proven that *L. acidophilus* consumption produces and upregulates neurotransmitter and neurotrophic factors, including GABA and serotonin (35–38). Thus, it is estimated that increased *L. acidophilus* EG004 in the gut modulates neurotransmitters and affects the nerve system of an animal. Second, SCFAs, fermentation products of *L. acidophilus*, apply positively to brain function. For example, acetate, one of the short-chain fatty acids (SCFAs), promotes the activation of the parasympathetic nervous system (39). Also, it is indicated that acetate improved cognitive ability and neurogenesis in the hippocampus with increasing BDNF and insulin-like growth factor 1 (IGF-1) levels as a glatiramer acetate form (40). Likewise, butyrate, a famous histone deacetylase (HDAC) inhibitor, has been used for pharmacological purposes since lower global histone acetylation is a

common phenomenon observed in many neurodegenerative diseases (41). Its therapeutic effect on neurodegenerative diseases, including Parkinson's disease, was verified, showing enhancement of neurotrophic factors and improvement in learning and memorizing (42). However, SCFAs are not produced until nondigestible carbohydrates reach the small intestine to be broken down by microbial metabolism, so it is not fully produced by the human digestive enzymes without specific microbes. *L. acidophilus* is a representative species that produces SCFAs through nondigestive carbohydrates, and it can be assumed that the intake of *L. acidophilus* EG004 caused the increase in SCFAs of the gut of experimental mice. The result of SCFA measurement in bacterial culture raises the possibility of this assumption (Table S2). Although it is different from the metabolism in the gut since the SCFAs were measured in the medium to which glucose is the main energy source, it estimates indirectly its SCFA-producing ability. The result of functional profiling in our study also upholds this conclusion (Fig. 5B). In the analysis of functional profiling, the activation of genes of synthesis and degradation of ketone bodies were predicted by comparison with those of the control. The ketone body is one of the main fuels of the brain like lactate and butyrate, which is the main product of *L. acidophilus*, and is also capable of replacing glucose as an alternative fuel. Similar to butyrate mentioned earlier, ketone bodies modulate the brain with an antioxidant reaction, energy supply, regulation of deacetylation activity, and regulation of the immune system. In recent studies, it is indicated that the increase of the ketone body concentration induces an alleviation effect on brain diseases, such as epilepsy, Alzheimer's disease, and Parkinson's disease as well as memory improvement (43–45). Based on this evidence, ingested *L. acidophilus* EG004 in our experimental group seems to have produced SCFAs and modulated neurotransmitters, and *L. acidophilus*-derived metabolites would have increased the cognitive ability of the host. Although we did not measure microbial-derived metabolites, previous research demonstrated that probiotic consumption leads to an increase of microbial-derived metabolites in the intestines.

Among detected bacterial species with the ratio difference, several of them were indicated as important factors in the research of brain disease. *Adlercreutzia equolifaciens* is a bacterium that produces equol (phytoestrogen), which obstructs microglial function. In previous studies, a higher ratio of *A. equolifaciens* was found in the gut of patients with Alzheimer's disease and autism spectrum disorder (46, 47). In other studies, *Roseburia hominis* and *Bacteroides_F pectinophilus* were detected with a higher ratio in the patients with Alzheimer's disease than that in healthy persons (48, 49). When comparing the gut microbiome between the Parkinson's disease group and normal group, we discovered *Soleaferrea massiliensis* more frequently in the patient group (50). Interestingly, the strains that showed a high ratio from the previous studies of brain disease patients were found to show a lower ratio in the *L. acidophilus* group than that in the control group (Fig. 4F). A decreased bacterial ratio related to brain diseases seems to positively affect cognitive ability, and we believe that it is due to *L. acidophilus* consumption. Although the specific mechanism cannot be estimated in this study, it seems to be influenced by the ingestion of *L. acidophilus* EG004. We hope that it will be a clue for unraveling the role of *L. acidophilus* in the gut-brain axis in further studies.

In a functional profiling analysis, we offered explainable factors for the microbial effect on the brain. Three KEGG categories were related to toxic chemical degradation, as follows: dioxin degradation, xylene degradation, and caprolactam degradation (Fig. 5B). Dioxin, a neurotoxin, has the risk for autism and neurodegenerative disease (51, 52). Xylene inhibits normal protein synthesis of neuronal function and induces instability in the neuronal membrane. When it is inhaled, psychological deficits can be caused (53, 54). These chemicals are noxious to the brain, so activation of these chemical degradations would have diminished negative effects in the *L. acidophilus* group. Besides, two KEGG categories related to the immune system were found. One of them is *Staphylococcus aureus* infection, which is known to cause brain abscess. Since there

have been many studies demonstrating that *L. acidophilus* has antimicrobial activity against *S. aureus*, activation of this category is thought to be due to an increase in the amount of *L. acidophilus*. The function of renal cell carcinoma was predicted in the experimental group. As it involves not only tumor suppressor genes, such as VHL, GH, and BHD, but also oncogenes, such as MET and PRCC-TFE3, it seems to be necessary to confirm the exact mechanism and side effects.

The purpose of this study was to develop a new strain that has positive effects on brain function, which can be recognized through changes in cognitive processes. Also, we aimed to provide an underlying biological mechanism by the gut microbiome that affects the brain. It is necessary to measure metabolite changes in order to provide an understanding of the mechanism of altered cognitive ability. However, the altered metabolite from the animal body was not identified fully. To overcome this limitation, we conducted the metagenome analysis, correlation analysis between cognitive ability and gut microbiome, measurement of SCFA-producing ability, and whole-genome comparison analysis. These analyses were not covered in the identification of a biological factor that caused improved cognitive ability but presented a group of genes and mechanisms that can infer the process. Although we did not provide direct evidence of phenotype changes caused by probiotic ingestion, we hope that our findings will help infer the process of the gut-brain axis.

## MATERIALS AND METHODS

**Animals.** Four-week-old C57BL/6 mice (*n* = 48; average weight, 26 g) were gained from YoungBio (Seongnam, South Korea). Since male mice have been used as animal models for the gut-brain axis and it is estimated that there is no difference between the intestinal environment and gut-brain axis system between females and males, male mice were used for the experiment to reduce the experimental variation. All mice were housed in a group of four per cage under standard controlled laboratory conditions (temperature of 20°C $\pm$ 5°C, humidity of 55%~60%), on a 12-h light/dark cycle [light on at 7:00 a.m.]). Each group consisted of 12 mice, and 4 mice were distributed to 3 cages. Twelve cages were located at random. All animals received *ad libitum* access to food. All animal experiments were performed following protocols approved by the Institutional Animal Care and Use Committee (IACUC) of Seoul National University, and the permission number is SNU-190607-4-3.

**Bacterial treatment.** The bacterial strains were isolated from fermented dairy foods. When identifying the gut-brain axis effect, we determined that the important factors to be considered were viability and adherence capacity. Therefore, we selected the species that are known to have adherence capacity in the GI tract, as well as the potential for a gut-brain axis effect. To identify species of each strain, 16S rRNA genes were sequenced by Macrogen Inc. (Seoul, South Korea) with 27F and 1492R primers. Obtained sequences were compared with sequences in the NCBI database using BLAST. The experiment consisted of 4 groups; 3 experimental groups were fed on autoclaved tap water mixed with *L. acidophilus* EG004, *L. paracasei* EG005, and *L. rhamnosus* EG006; and a control group was fed on sterilized tap water. Each group consisted of 12 mice. Bacteria for delivery to mice were cultivated freshly every day. Probiotic colonies were subcultured into 5 mL MRS broth for 8 hours. After the subculture, 3 probiotic strains were inoculated in 500 mL MRS broth for 16 hours. Cultivated cells were centrifuged at 4,000 rpm for 10 min. The supernatant was removed, and the pellet was suspended in a 0.85% NaCl solution. Resuspended cells were centrifuged at 4,000 rpm for 10 min to remove medium ingredients. The washing process was conducted twice. Washed cells were dissolved into autoclaved tap water. The final cell concentration was about 1.0E9 CFU/mL. To estimate the probiotic amount per day per subject, the daily water intake and probiotic concentration were recorded. The cell viability of probiotics was measured by serial dilution and spreading onto an MRS agar plate. The probiotic amount per day per subject was calculated as an average of daily water intake per subject, by multiplying the average of daily probiotic concentration.

**Animal treatment.** The animal experiment was designed to minimize animal stress. All animal treatments are described in Fig. 1 in a timeline. Four-week-old mice were allowed to habituate freely for acclimatization for 1 week. After a week, tap water and water mixed with probiotics were delivered every day. Water intake was monitored every day and body weight was measured every week. Evaluations of cognitive ability were conducted after 4 weeks after probiotic intake. Behavioral tests were conducted at least 2 days after the weight measurement day to minimize the stress effect. Animals were carried to a behavioral test room to assimilate room conditions and were allowed to relax for 6 hours before any behavioral test. In order to reduce the variance of feeding time, the experimental order of the mice was distributed evenly. All apparatus and objects for the behavioral tests were cleaned with 70% ethanol and dried after every trial to remove odors and any clues. The mice were sacrificed at the end of 13 weeks after the evaluations of the cognitive behavior. Preliminary experiments were conducted to obtain appropriate experimental values under our experimental environmental conditions. The three to five experimental conditions referring to published results were tested in our laboratory, and the experimental conditions showing a value similar to the average value of the previous studies were determined.

**Y maze (spontaneous alternation).** Short-term spatial memory was assessed with a Y maze apparatus. Spontaneous alternation (SA) was used to measure the habit of rodents to explore a new environment. The Y maze consisted of 3 identical arms that cross each other with 120° (JeungDo Bio & Plant, Co., Ltd., South Korea). Mice were laid in the middle of the Y maze facing a corner, not an arm. Each animal was allowed to freely navigate all three arms for 5 minutes, and the entries to any arm were recorded. An arm entry was determined as any instance when the whole body of the mouse entered the arm and navigated at least 70% of the space. The spatial memory was evaluated by spontaneous alternation, the number of arm entries, and the ratio of mice per group that entered spontaneous alternation during the first three entries. Spontaneous alternation was calculated as shown below.

$$\text{spontaneous alternation } (\%) = \frac{\text{number of spontaneous alternation}}{\text{total number of arm entries} - 2} \times 100$$

**Novel object recognition test (NOR).** Based on the concept that mice tend to prefer a new object over a familiar one, a novel object recognition test (NOR test) was performed in an open field (40 by 40 by 40 cm [width by depth by height]; JeungDo Bio & Plant Co., Ltd., South Korea). Two objects for this test were selected showing similar preferences through the preference test. The test consisted of sample trial (T1; 10 min), interval time (IT; 60 min), and novel object trial (T2; 5 min). In T1, 2 identical objects were located at the one-third and two-thirds diagonal of the open field, respectively. The animal was laid facing the wall with the same distance to two objects and was allowed to explore objects for 10 min. After exploration, the mouse came back to the cage and had a rest. In T2, objects were positioned at the same position as T1, but one of the objects was changed to a novel object. To measure the time taken to interact with objects, all experiment processes were recorded, and the exploration time was measured by Movavi software with 3 decimal places. It was recognized as significant only when the mouse approached facing the objects within 2.5 cm. Cases in which the mouse climbed objects and individuals with an exploration time of less than 2 seconds were excluded. The results were presented as a discrimination ratio, the number of object touches, and the ratio of a mouse that touched the novel object first before it touched the familiar object. The discrimination ratio was defined as the following equation.

$$\text{discrimination ratio } (\%) = \frac{\text{novel object interaction time}}{\text{novel object interaction time} + \text{familar object interaction time}} \times 100$$

**Passive avoidance task (PAT).** The passive avoidance task is designed to evaluate inhibitory avoidance memory according to the rodent habit that a mouse prefers a dark environment naturally. The shuttle box (41 by 21 by 30 cm [width by depth by height]; JeungDo Bio & Plant Co., Ltd., South Korea) is an apparatus made for the passive avoidance task and consists of a bright chamber and a dark chamber which are separated by a sliding door. The floor of the chambers is made of stainless-steel grids to flow current. The test was conducted for 2 days for acquisition (day 1) and test (day 2). On day 1, a subject was put in the bright chamber facing the wall across the closed sliding door. After the mouse explored the bright chamber for 1 minute, and the moment the mouse was away from the door for over 100 mm, facing the wall (not the door), the door was opened so that the mouse could freely enter and move around the dark chamber. Latency time was measured until the mouse entered the dark chamber completely. The door was closed when the animal entered the dark compartment, including its tail, and a 0.25-mA electric shock was provided to the paws by a steel grid for 3 seconds. To memorize the situation, the mouse was kept in the dark chamber for 30 seconds after the shock and returned to the home cage for 24 hours. On day 2, the mouse was laid again into the bright chamber. After 1 minute of adaptation, the sliding door was opened when the mouse faced the wall like on day 1. Latency time was measured again until the mouse entered the dark chamber. If the animal stayed in the bright chamber for more than 300 seconds (which was the cutoff time), the experiment was completed. All experimental processes were recorded, and the time was measured by the Movavi program with 3 decimal places.

**Y maze (forced alternation).** Forced alternation (FA) was assessed with the same Y maze as described above. This test consisted of 3 phases, including a training trial (T1; 5 min), interval time (IT; 60 min), and test trial (T2; 5 min). A mouse was placed at a starting arm of the Y maze facing the wall. The subject freely explored the maze during T1, while an entry was blocked with white expanded polystyrene. After the learning trial, the mouse was returned to the home cage and rested for 1 hour. In T2, the mouse was again placed into the starting arm without the plate blocking the novel entry, and it explored all three arms. All movements of mice were recorded through video recordings. Forced alternation was evaluated by the ratio of time spent in the novel arm compared with that of the whole experimental time, time taken to first enter the novel arm, and the percentage of mice per group that entered the novel arm as their first entry. A case in which the mouse explored at two-thirds of the arms was admitted as a valid entrance. An individual that showed no navigation of the maze or that had entered the arms less than 5 times was excluded.

**Feces collection and cognitive ability evolution.** After all cognitive assessments had been completed, 2 to 3 stool samples were taken from each experimental subject. Sterilized stainless-steel tweezers were used for fecal picking; tweezers were washed with 70% alcohol and dried sufficiently before collecting new samples. The fresh samples were enclosed immediately into a 1.5-mL Eppendorf tube and were put on ice. Then, samples were stored at −80°C until used for 16S rRNA sequencing.

In order to determine the group that showed the best increase in cognitive ability, a score was assigned to the cognitive ability evaluation item. The items used for evaluation were spontaneous

alternation, group ratio of SA, discrimination ratio, group ratio of NOR, step latency at day 2, forced alternation, and group ratio of FA (Table S2). Scores were given in ascending order of ranking (1 to 4 points), and the group with the highest total was selected as the group with the highest cognitive ability increase.

**Statistics.** Data were analyzed by R studio. Ineligible data were cut based on the requirements mentioned above. Data normality was assessed using the Shapiro-Wilks test, and homogeneity of variance was assessed using Levene's test. Wilcoxon signed-rank test and independence *t* test was used to evaluate statistical significance between experimental groups. *P* values were adjusted by the false discovery rate (FDR) method for multiple testing corrections. Statistical significance was set as a *P* value under 0.05. All data are expressed as mean ± SEM.

**Full 16S-23S rRNA sequencing.** To characterize the microbial community associated with measured cognitive assessment, metagenome sequencing of the 16S-23S rRNA gene was carried out by using an Oxford Nanopore MinION device. Metagenome sequencing was performed for the control group and *L. acidophilus* group, which showed a significant difference from the control in the cognitive ability evaluation. Among the 12 stored stool samples of each group, 5 samples with a sufficient amount for sequencing were selected. For library construction, genomic DNA (gDNA) was extracted from fecal samples using the AccuPrep stool DNA extraction kit (Bioneer, Daejeon, South Korea). To identify the quality of extracted gDNA, $A_{260}/A_{280}$ and $A_{260}/A_{230}$ were used with 0.7% agarose gel electrophoresis. After a quality control was performed, selected samples were used for the library construction. Stool samples were lysed. and bacterial cells were disrupted by zirconia/silica beads and proteinase K. The sequencing library was prepared by 16S-26S rRNA PCR amplification with a ligation kit (SQK-LSK109; Nanopore, Oxford, UK) following the manufacturer's instructions. Purification and quality checks were conducted using the Agencourt AMPure XP cleanup kit (Beckman Coulter, CA, USA), Quant-iT PicoGreen double-stranded DNA (dsDNA) assay kit (Invitrogen, Ireland), and 0.7% agarose gel. The PCR products were diluted and end repaired using NEBNext formalin-fixed, paraffin-embedded (FFPE) repair mix (New England BioLabs, Ipswich, USA). The amplicon was nick repaired using an NEBNext end repair/dA-tailing module (New England BioLabs), prior to adapter ligation by NEBNext quick ligation module (New England BioLabs). The sequencing library was loaded on a primed Flongle flow cell according to the Nanopore protocol. Sequencing was performed on a MinION MK1b device. Sequencing data were acquired by MinKNOW software (19.12.5) without live base calling.

**Metagenome analysis.** Raw data were obtained as fast5 files. Base calling was carried out by Guppy 4.0.11 with 2,000 chunk size and 4 base callers (55). Porechop v3 was executed for trimming adapter sequences (https://github.com/rrwick/Porechop). To annotate bacterial taxonomy, trimmed sequences were aligned with MIrROR (http://mirror.egnome.co.kr/) using Minimap2 (56). In OTU identification, only results with more than 2,500 matching bases and more than 3,500 bases, including gaps in mapping, were used. To normalize abundance data, the trimmed mean of M-values (TMM) method was used with the edgeR package of R software (57). To characterize each group, biological diversity was calculated through the physeq package of R software (58). A rarefaction curve was constructed to check the saturation of genome sequencing. To compare species richness, alpha diversity was calculated as chao1 and Shannon indexes. To compare between groups, beta diversity was calculated using Bray-Curtis dissimilarity and Unifrac distance. The *P* value was calculated by using the Adonis test. For the detection of unequal features, a Wilcoxon signed-rank test was performed in each taxonomic level with a 0.95 confidence level. To compare the functional profile, PICRUSt2 was used (59). Correlation between cognitive ability and bacterial OTUs was inferred by Spearman's rank correlation analysis. *P* values were adjusted by using the FDR method.

**SCFA identification in bacterial culture.** To identify the amount of short-chain fatty acids (SCFAs), high-performance liquid chromatography (HPLC) was performed using the Ultimate3000 system (Thermo Dionex, USA) and the Aminex 87H column (300 by 10mm; Bio-Rad, USA). Bacterial cultures of EG004, EG005, and EG006 were inoculated for 24 hours. After cultivation, the samples were filtered with 0.45 $\mu$m of a membrane filter. The filtered sample of 10 $\mu$L was injected into the HPLC instrument.

**Whole-genome sequencing of EG005 and EG006 and whole-genome sequence of EG004.** To identify probiotic safety and probiotic potential secondary metabolite-producing ability, whole-genome sequencing of *L. paracasei* EG005 and *L. rhamnosus* EG006 was performed. For library construction, DNA was extracted from cultured bacterial cells. After a quality control was performed, gDNA was used for the library construction. Bacterial cells were lysed by lysozyme for Gram-positive bacteria, and RNA and proteins were removed to isolate DNA. Quality control for gDNA was conducted by using $A_{260}/A_{280}$ and $A_{260}/A_{230}$ with a 0.8% agarose gel. Genomic DNA was fragmented to a target length of 20 kb using g-Tube (Covaris, MA, USA), and short DNA fragments of <5 kb long were depleted by Short Read Eliminator (SRE) (Circulomics, MD, USA). The fragments were end repaired, nick repaired, and then ligated with a Nanopore adapter. After every enzyme reaction, the DNA samples were purified using AMPure XP beads (Beckman Coulter, CA, USA) and quality controlled (QC) with a Quant-iT PicoGreen dsDNA assay kit. The sequencing library was loaded on a primed Flongle flow cell according to the Nanopore protocol. Sequencing was performed on a MinION device by using MinKNOW software.

Base calling from raw data was conducted by Guppy Basecaller v4.0.15 with filtering with an average base call Phred quality score. Adapter sequences were trimmed by PoreChop v0.2.4. Genome assembly was conducted by Canu. Assembled contigs were polished by Nanopolish, racon, and pilon. Circlator circularized each contig and detected replication origin. The assembled contig was assessed by BUSCO 3.0.2. The complete sequence of *L. acidophilus* EG004 that is deposited in the NCBI database with accession number PRJNA657145 was used (60).

**Comparative analysis of bacterial genome sequences.** A genetic map was generated by the CGView server (61). To check the safety and functionality of bacteria as probiotics, genetic factors were identified by whole-genome sequences. A virulence factor and prophage gene were detected by VirulenceFinder 2.0 and PHASTER, respectively. IslandViewer4 identified a genomic island and crisprfinder searched the CRISPR region. Bacteriocin detection was conducted by BAGLE4. To compare functional gene contents, protein prediction was performed by the RAST server. Predicted protein sequences were classified by the SEED system. Categorized protein sequences are showed as the proportion in the total predicted sequences.

**Data availability.** The complete sequences of *L. paracasei* EG005 and *L. rhamnosus* EG006 are available in the NCBI database with accession numbers, SAMN23227569 and SAMN23227570, respectively. The metagenomic sequences are available in the NCBI database under the accession number PRJNA781018.

## SUPPLEMENTAL MATERIAL

Supplemental material is available online only.
**SUPPLEMENTAL FILE 1**, PDF file, 0.9 MB.

## ACKNOWLEDGMENTS

This research was supported by the eGnome, Inc. (Republic of Korea).

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
