## [Reviewer comments · Microbiology Spectrum]

Microbiology Spectrum

Positive effect on cognitive ability of *Lactobacillus acidophilus* EG004 in healthy mouse and fecal microbiome analysis using full-length 16S-23S rRNA metagenome sequencing

Soomin Jeon, Hyaekang Kim, Jina Kim, Jinchul Jo, Donghyeok Seol, Youngseok Choi, Seoae Cho, and Heebal Kim

Corresponding Author(s): Heebal Kim, Seoul National University

Review Timeline:

Submission Date:	October 20, 2021
Editorial Decision:	November 15, 2021
Revision Received:	November 21, 2021
Editorial Decision:	November 29, 2021
Revision Received:	December 6, 2021
Accepted:	December 7, 2021

Editor: Jan Claesen

Reviewer(s): Disclosure of reviewer identity is with reference to reviewer comments included in decision letter(s). The following individuals involved in review of your submission have agreed to reveal their identity: Bo Cui (Reviewer #2); Rustem Abuzarovich Ilyasov (Reviewer #3)

Transaction Report:

DOI: <https://doi.org/10.1128/Spectrum.01815-21>

November 15, 2021

Prof. Heebal Kim
Seoul National University
Seoul
Korea (South), Republic of

Re: Spectrum01815-21 (Positive effect on cognitive ability of *Lactobacillus acidophilus* EG004 in healthy mouse and fecal microbiome analysis using full-length 16S-23S rRNA metagenome sequencing)

Dear Prof. Heebal Kim:

Thank you for submitting your manuscript to Microbiology Spectrum. As you will see your paper is very close to acceptance. Please modify the manuscript along the lines I have recommended. As these revisions are quite minor, I expect that you should be able to turn in the revised paper in less than 30 days, if not sooner. If your manuscript was reviewed, you will find the reviewers' comments below.

When submitting the revised version of your paper, please provide (1) point-by-point responses to the issues I raised in your cover letter, and (2) a PDF file that indicates the changes from the original submission (by highlighting or underlining the changes) as file type "Marked Up Manuscript - For Review Only". Please use this link to submit your revised manuscript. Detailed instructions on submitting your revised paper are below.

Link Not Available

Sincerely,

Jan Claesen

Reviewer comments:

Reviewer #3 (Public repository details (Required)):

The metagenome sequencing data (16S-23S rRNA) should be submitted to GenBank if it is not submitted yet.

Reviewer #3 (Comments for the Author):

Manuscript: Spectrum01815-21 Positive effect on cognitive ability of *Lactobacillus acidophilus* EG004 in healthy mouse and fecal microbiome analysis using full-length 16S-23S rRNA metagenome sequencing

The aim of the paper is to study effect on cognitive ability of *Lactobacillus acidophilus* EG004 in healthy mouse and fecal microbiome analysis using full-length 16S-23S rRNA metagenome sequencing. In the manuscript, the authors studied a bacterial strain *Lactobacillus acidophilus* EG004 with a positive effect on cognitive ability using a healthy animal model. The authors experimentally verified improved cognitive ability by cognitive behavioral tests. The authors performed full 16S-23S rRNA sequencing and provided gut microbiome composition at a species level. The provided microbiome composition consisted of candidate microbial groups as a biomarker that shows positive effects on cognitive ability. Therefore, their study suggests a new perspective for probiotic strain use applicable for medicine. The uniqueness of the text is 90% by AntyPlagiarism.net. The manuscript is written well. English is proper, well understandable.

Reviewer has some comments:

Line 74 - most of researches were - should be - most of the researches was.
Line 73 - many researches - should be - many pieces of research.
Line 82 - industrialization process - should be - industrialization processes.
Line 105 - for the sentence - Autism, Alzheimer's disease, and Parkinson's disease (7-9) - add additional citation (Danilenko et al., 2021) and add to the References - Danilenko, V.N., Devyatkin, A.V., Marsova, M.V., Shibilova, M.U., Ilyasov, R.A., Shmyrev, V.I., 2021b. Common inflammatory mechanisms in COVID-19 and Parkinson's diseases: the role of microbiome and probiotics in their prevention. Journal of Inflammation Research 14, (In press). doi: 10.2147/JIR.S333887.
Line 108 -to the sentence - the neural pathways of the brain-gut axis (10). - add additional citation (Fetissov et al., 2019). and add to the References - Fetissov, S.O., Averina, O.V., Danilenko, V.N., 2019. Neuropeptides in the microbiota-brain axis and feeding behavior in autism spectrum disorder. Nutrition 61, 43-48. doi: 10.1016/j.nut.2018.10.030.
Line 112 - Second, the second suggestion - should be - Second, the suggestion
Line 113 - microbiome affect brain - should be - microbiome affects brain.
Line 113 - metabolic pathway - should be - metabolic pathways.
Line 127 - remove one dot.
Line 153 - The averages daily - should be - The averages of daily.
Line 168 - In the comparison of - should be - The comparison of.
Line 194 - light room - should be - lightroom.
Line 195 - remove italics of the word - group.
Line 226 - comparison - should be - comparative.
Line 236 - familiae - should be - families.
Line 275 - whole genome - should be - whole-genome.
Line 308 - recognition test and passive avoidance task - should be - recognition tests and passive avoidance tasks.
Line 321 - were - should be - was.
Line 343 - factor - should be - factors.
Line 350 - purpose - should be - purposes.
Line 370 - these evidences - should be - this evidence.
Line 398 - negative effect - should be - negative effects.
Line 408 - to provide - should be - provide.
Line 413 - These analyses were not covered to identification of a biological factor caused - should be - These analyses were not covered in the identification of a biological factor that caused.
Line 416 - probiotics ingestion - should be - probiotic ingestion.
Line 444 - by - should be - at.
Line 442 - with - should be - at.
Line 453 - add space after dot.
Line 457 - from probiotics intake - should be - after probiotic intake.
Line 459 - room condition - should be - room conditions.
Line 472 - rodent's habit - should be - rodents' habits.
Line 478 - entered - should be - that entered.
Line 486 - preference - should be - preferences.
Line 516 - After 1 minute for adaptation - should be - After 1 minute of adaptation.
Line 531 - time taken - should be - time is taken.
Line 554 - correction - should be - corrections.
Line 789 - statistic - should be - statistics.
The metagenome sequencing data (16S-23S rRNA) should be submitted to GenBank.
Please check English by professional translator one more times.
In further authors should study details of biological factors and molecular mechanisms that caused improved cognitive ability in mice after treatment with *L. acidophilus* EG004 strain.

No other comments.

A minor revision is required.

Preparing Revision Guidelines

- point-by-point responses to the issues I raised in your cover letter
- Upload a compare copy of the manuscript (without figures) as a "Marked-Up Manuscript" file.
- Each figure must be uploaded as a separate file, and any multipanel figures must be assembled into one file.

- Manuscript: A .DOC version of the revised manuscript
- Figures: Editable, high-resolution, individual figure files are required at revision, TIFF or EPS files are preferred

Please return the manuscript within 60 days; if you cannot complete the modification within this time period, please contact me. If you do not wish to modify the manuscript and prefer to submit it to another journal, please notify me of your decision immediately so that the manuscript may be formally withdrawn from consideration by Microbiology Spectrum.

**Title**

Positive effect on cognitive ability of *Lactobacillus acidophilus* EG004 in healthy
mouse and fecal microbiome analysis using full-length 16S-23S rRNA
metagenome sequencing

**Running title**

Positive effect on cognitive ability of *Lactobacillus acidophilus* EG004

**Authors**

Soomin Jeon^{a,†}, Hyaekang Kim^{a,†}, Jina Kim^a, Donghyeok Seol^{a,b}, Jinchul Jo^a,
Youngseok Choi^a, Seoae Cho^b, Heebal Kim^{a,b,d,*}

12 ^aDepartment of Agricultural Biotechnology and Research Institute of Agriculture
and Life Sciences, Seoul National University, Seoul, Republic of Korea

14 ^beGnome, Inc, Seoul, Republic of Korea

15 ^dInterdisciplinary Program in Bioinformatics, Seoul National University, Seoul,
16 Republic of Korea

17 ^{*}Corresponding authors

† Soomin Jeon and Hyaekang Kim contributed equally to this work. Author order
was determined retroalphabetically.

Soomin Jeon

Email: soty23@snu.ac.kr

Hyaekang Kim

Email:hkim458@snu.ac.kr

Jina Kim

Email:jinak750@gmail.com

Donghyeok Seol

Email:sdh1621@snu.ac.kr

Jinchul Jo

Email: macjoo2000@snu.ac.kr

Youngseok Choi

Email:seok1213neo@snu.ac.kr

Seoae Cho

Email:seoae@egenome.co.kr

Heebal Kim

Email: heebal@snu.ac.kr

*Correspondence

Heebal Kim, Department of Agricultural Biotechnology and Research Institute of
Agriculture and Life Sciences, Seoul National University, Seoul, Republic of
Korea

Email: heebal@snu.ac.kr

Tel.: +82-2-880-4822

Fax: +82-2-883-8812

**Word count**

Abstract; 249 words

Text; 4245 words (excluding materials and methods) and 6903 words (including
materials and methods)

**Abstract**

The concept of the ‘Gut-brain axis’ has risen. Many types of research demonstrated the effect and
mechanism of the GBA. Although many studies have been reported, most of the studies are
focused on neurodegenerative disease and it is still not clear what type of bacterial strains have
positive effects on the brain. Therefore, we designed an experiment to discover a strain that
positively affects cognitive ability using healthy mice. The experimental group consisted of a
control group and three probiotic consumption groups, *Lactobacillus acidophilus*,
*Lacticaseibacillus paracasei*, and *Lacticaseibacillus rhamnosus*, which are verified to have
beneficial effects for host health as the gut microbiome. Cognitive ability was measured by 4
cognitive-behavioral tests and the group fed on *L. acidophilus* showed the most improved
cognitive ability. To provide an understanding of the altered microbial composition effect on the
brain, we performed full 16S-23S rRNA sequencing using Nanopore, and OTUs were identified
at a species level. In the group fed on *L. acidophilus*, the intestinal bacterial ratio of Firmicutes
and Proteobacteria phyla increased and the bacterial proportions of 16 species were significantly
different from those of the control group. We estimated that the positive results on the cognitive
behavioral tests were due to the increased proportion of *L. acidophilus* EG004 strain in the
subjects’ intestines since the strain is capable of producing butyrate and therefore modulating
neurotransmitters and neurotrophic factors. We expect that our new strain expands the industrial
field of *L. acidophilus* and helps understand the mechanism of the brain-gut axis.

**Importance**

In recent, the concept of 'gut-brain axis' has risen that microbes in the GI tract affect brain by

modulating signal molecules. Although many researches were reported in a short period, a
signaling mechanism and effect of a specific bacterial strain are still unclear. Besides, since most
of researches were focused on neurodegenerative disease, the study with a healthy animal model
is still insufficient. In this study, we provide a bacterial strain (*Lactobacillus acidophilus* EG004)
with a positive effect on cognitive ability using a healthy animal model. We experimentally
verified improved cognitive ability by cognitive behavioral tests. We performed full 16S-23S
rRNA sequencing using nanopore MinION, and provided gut microbiome composition at a
species level. The provided microbiome composition consisted of candidate microbial groups as
a biomarker that shows positive effects on cognitive ability. Therefore, our study suggests a new
perspective for probiotic strain use applicable for various industrialization process.

**Keywords**

*Lactobacillus acidophilus*, gut microbiome, gut-brain axis, cognitive ability, Nanopore
sequencing

**Introduction**

The human body is a complex community that habituates various bacteria. Among the
bacterial communities in the human body, the gastrointestinal tract is the best bacterial
community that has the most abundant and various bacteria (1). In 2006, having been released
research that obesity is associated with bacterial composition in the gut, a study for gut
microbiome began in earnest (2). The gut microbiome is defined as the collective genomes of
microorganisms that live in the gastrointestinal tract. Functions of the gut microbiome have been
reported such as nutrient metabolism and regulation of the immune system for the host (3).
Microbial composition in the gut is altered by environmental factors like age, diet, stress, and
lifestyle, and the change in microbial composition can induce physical changes in the host (4). In
recent, the gut microbiome's effects on the brain have been proved and the concept of the brain-
gut axis has risen to the surface (5). The brain-gut axis is a complex system involving the enteric
nervous system and central nervous system including the brain and spinal cord, and it works with
bidirectional communication between the central and the enteric nervous system (6). Although
the brain is located apart from the gut, the gut microbiome can affect the brain by stimulating the
enteric nervous system and vagus nerve. Thus, dysbiosis of the gut microbiome often causes
brain diseases. The recent experimental results described that gut microbiome dysbiosis was
observed in patients with **Autism, Alzheimer's disease, and Parkinson's disease (7-9)**. At the
same time, studies on the mechanisms to understand the brain-gut axis have been conducted.
First, it was suggested that the microbial-derived metabolites are the main components acting on
**the neural pathways of the brain-gut axis (10)**. The most well-studied substances are short-chain
fatty acids (SCFA) such as acetate, propionate, and butyrate, which are produced in the process
of decomposing non-digestible fibers and carbohydrates (11). It promotes indirect signaling to

the brain by modulation and induction of neurotransmitter and neurotrophic factors like γ -
aminobutyric acid (GABA) and Brain-derived neurotrophic factor (BDNF). Second, the second
suggestion was that the gut microbiome affect brain function by regulating metabolic pathway
(12). Previous research reported that the level of tryptophan metabolites including serotonin and
indolepyruvate was altered by the gut microbiome. These metabolites have roles in the
functioning of the gut-brain axis such as signaling and anti-oxidant. Third, the gut microbiome
may affect the brain by immune pathway (13). Interferon (IFN), Tumor necrosis factor (TNF),
and Interleukin are well-known immune factors. According to recent studies, the amount of the
immune factors is regulated by the intestinal microflora. These immune factors affect brain
function by stimulating and activating the hypothalamic-pituitary-adrenal axis. Finally, it was
suggested that gut microbes directly influence the brain by altering the fatty acid composition of
the brain (14). Several studies have been reported on the correlation between intestinal
microbes and the brain, but the specific mechanism of the brain-gut axis is still not clear.

Probiotics are defined as bacteria that have positive effects on the host body (15). Probiotics
have been widely used as a health supplement since it has various beneficial functions to host's
health with high adhesion property to the intestine and low side effect. Most probiotics include
bacteria genera that are gram-positive, facultative anaerobic and rod-shaped.. *Lacticaseibacillus*
*rhamnosus* (*Lcb. rhamnosus*) is one of the longest-studied probiotic species, and many strains
such as LGG and GR-1 belonging to this genus are commercially available. It is well known that
*Lcb. rhamnosus* has positive effects on diarrhea, acute gastroenteritis, and atopic dermatitis (16-
18). Recently, its neurobehavioral effects such as anxiety and depression relief have been
reported (19). *Lacticaseibacillus paracasei* (*Lcb. paracasei*) is one of the representative probiotic
species, and it has been studied to be effective in treating ulcerative colitis and allergic rhinitis

(20, 21). In a recent study, an effect on age-related cognitive decline and a stress relief effect was
reported with several strains of this species (22). *Lactobacillus acidophilus* (*L. acidophilus*) is
another representative probiotic strain. This strain lowers cholesterol levels and has beneficial
health effects such as antibacterial effects against harmful bacteria like *Streptococcus mutans* and
*Salmonella typhimurium* (23, 24).

In this study, we aimed to present a new strain that has an enhancing effect on cognitive
ability through the brain-gut axis and provide an additional understanding of the brain-gut axis.
Three probiotic strains, *L. acidophilus*, *Lcb. paracasei*, and *Lcb. rhamnosus*, which have
previously demonstrated beneficial effects to the host as one of the gut-microbiome strains, were
used to confirm their positive effects on cognitive ability. Full 16S and 23S rRNA sequencing
was performed to annotate the gut microbiome at a species level for downstream analysis. We
expect our results to provide an understanding of the role of the gut microbiome.

**Results**

**Bacterial and animal treatments**

Three probiotic strains, *L. acidophilus* EG004, *Lcb. paracasei* EG005, and *Lcb. rhamnosus*
EG006, have been identified by the molecular method. These strains were clustered with
available *L. acidophilus*, *Lcb. paracasei*, and *Lcb. rhamnosus* strains, respectively, in a
phylogenetic tree of 16S rRNA gene (Figure S1). Probiotic strains were consumed by mice for 8
153 weeks with assessments of cognitive ability (Figure 1). The averages daily water intake per
154 subject were similar between groups (Figure 2A). Daily probiotic intakes were maintained
constantly and the average amount of *L. acidophilus* group, *Lcb. paracasei* group, and *Lcb.*
*rhamnosus* group were calculated as $(7.82E09 \pm 1.95E09)$, $(4.37E10 \pm 5.17E09)$, and
$(3.74E10 \pm 3.98E09)$ CFUs (Figure 2B). To identify the additional effect of probiotics, the body
weights of mice were measured every week (Figure 2C and S2). Patterns of weight gain in the 4
groups were similar for 8 weeks. The mean body weight gains of the control group showed the
highest value, which was 9.08 g. *Lcb. paracasei* group showed a significant difference from the
control group with P-value under 0.05 in the second measurement, but the difference was
immediately recovered. Similar to weekly weight change, statistical significance was not found
in accumulated weight between experimental groups for 8 weeks.

**Cognitive behavioral tests**

Spontaneous alternation test was conducted to assess spatial learning and short-term
memory. Although the average number of the total entries to each arm in *Lcb. paracasei* group

was slightly low, the difference between groups was not found (Figure 3A). In the comparison of
the mouse ratio showed spontaneous alternation for the first 3 entries, *L. acidophilus* group
showed the highest value as 75.0%. (Table S1). In spontaneous alternation, the average values of
probiotics-fed groups were higher than the value of the vehicle-fed group (Figure 3B). Among
the 4 experimental groups, *L. acidophilus* group showed the highest alternation ratio. Wilcoxon
rank-sum test was performed to identify statistical significance, but there was no statistical
difference between the experimental groups and control group.

Novel object recognition (NOR) test was performed to evaluate long-term and explicit
memory using 4 different features (Figure 3C, 3D, and Table S1). *L. acidophilus* group exhibited
the highest average ratio of mouse that touched the novel object before the familiar object,
whereas *Lcb. rhamnosus* group showed the lowest value under the control group. At
discrimination ratio comparison, the three probiotics-fed groups showed higher average values
than the control, and *L. acidophilus* group showed the highest values. To identify if there is a
significant difference, Wilcoxon rank-sum test was performed. When compared to the vehicle-
fed group, *L. acidophilus* and *Lcb. Paracasei* groups displayed statistically significant
differences with the adjusted P-value of 0.037. To identify animal behavior detail, the number of
objects touch and the total time of object observation in each group were compared. In a
comparison of object touch, statistical differences were significant in *L. acidophilus* and *Lcb.*
*Paracasei* groups with P-values of 0.031 and 0.042, respectively. Also, *L. acidophilus* group had
a significant difference between the time taken to observe the familiar object and the novel object.

Passive avoidance task was conducted to measure long-term and implicit memory. Step-
through latency was used to compare the mean difference between the experimental groups.

Most of the subjects were transferred into a darkroom for a minute on day 1 (Figure 3E). Only 3
animals took over 100 seconds to get into the darkroom. The difference between the
experimental group and the control was not found on day 1. When compared to the latency time
on day 1, the average latency time increased on day 2, and unexpectedly, 26 animals stayed in
the light room for over 300 seconds (Figure 3F). *Lcb. rhamnosus* group presented the highest
average latency time, followed by *L. acidophilus* group while the control group showed the
lowest average (Table S1). The Mann-Whitney U test was conducted to check the mean
difference, the P-values of *L. acidophilus* and *Lcb. rhamnosus* groups were less than 0.05
compared to the control group. The adjusted P values of both groups were 0.040.

To assess spatial learning and long-term memory, forced alternation was conducted.
Memory was evaluated by forced alternation (%), the number of arms that the mouse entered,
and the percentage of mice in a group that entered the novel arm as their first entry. While the
total number of the entries into each arm was diverse, there was no significant difference
between the experimental groups and control (Figure 3G). *L. acidophilus* group scored the
highest ratio of mice entered the novel arm as their first entry (Table S1). Forced alternation
values of *L. acidophilus* and *Lcb. rhamnosus* groups were higher than the value of the control
group (Figure 3H). Forced alternation of *Lcb. rhamnosus* group and the control group had a
significant difference with the adjusted P-value of 0.038.

**Full 16S-23S rRNA sequencing and biological diversity**

Metagenome sequencing was performed with *L. acidophilus* and control groups, which
showed the most improvement in cognitive ability. We compared the microbial composition of

both groups. Gut microbial component information annotated at a species level was completely
constructed by sequencing the entire 16S-23S rRNA of the mouse stool (Table 1). Averagely,
323870.0±84085.5 reads were generated from 10 stool samples. The total number of identified
OTU was 252401.6±56284.7 in *L. acidophilus* group and 259945.6±78526.0 in the control group.
The produced OTUs were annotated as a total of 528.4±90.4 species in *L. acidophilus* group and
539.8±55.4 species in the control group. To check the sufficiency of the sequencing depth for the
analysis, a rarefaction curve was created (Figure 4A).

Alpha diversity was calculated to compare species richness within a group (Figure 4B). In
the comparison of the two groups, no significant difference was found in Chao1 Shannon indexes.
Beta diversity was measured to compare the diversity of the microbial community between the
two groups (Figure 4C and D). It was confirmed that both beta diversity evaluations (Bray-Curtis
and Unifrac distance) had significant differences.

**Microbial composition**

In the comparison analysis of microbial compositions, taxonomies with significantly
different ratios were found between *L. acidophilus* group and the control group. At the phylum
level, Bacteroidota accounted for the highest proportion in both groups, followed by Firmicutes
(Figure 4E). Significant differences between the two groups were found in 2 of the 12 phyla
(Firmicutes, Proteobacteria), all of which were high in *L. acidophilus* group. At the class level,
Bacteroidia showed the highest proportion in both groups. Also, the proportion of Bacilli and
Gammaproteobacteria classes were increased in *L. acidophilus* group when compared to the
control group (Figure S3). At the order level, Bacteroidales showed the highest percentage in

both groups, and Lactobacillales and Enterobacterales orders were found to exhibit higher
proportions in *L. acidophilus* group. At the family level, *Muribaculaceae* showed the highest
proportion in both groups. It was found that 2 **familiae** (*Lactobacillaceae* and
*Enterobacteriaceae*) showed increased proportions in *L. acidophilus* group, while a decreased
percentage was observed in one family (*Ruminococcaceae*). In the Genus comparison,
*Muribaculum* genus showed the highest ratio in the two groups, and 12 genera showed
differences between groups. Three genera showed an increased proportion in the experimental
group, whereas 9 genera showed higher mean values in the control group. The genus increased in
*L. acidophilus* group were *Lactobacillus*, *Staphylococcus_A*, and *Escherichia*, whereas the
genera decreased in *L. acidophilus* group were *Bacteroides_F*, *Desulfotomaculum*,
*Lachnobacterium*, *Bittarella*, *Agathobacter*, *Roseburia*, *Bariatricus*, and *Lachnospirarea*. At the
Species level, *Muribaculum intestinale* was found to account for the largest proportion, with over
50% in both groups. Following *M. intestinale*, the species *Lactobacillus acidophilus*,
*Lactobacillus johnsonii*, *Lactobacillus_B murinus*, and *Lactobacillus_H reuteri* were found with
a high proportion in *L. acidophilus* group, while *Lactobacillus_B murinus*, *Bacteroides_B*
*vulgatus*, *Faecalibaculum rodentium*, and *Kineothrix alysoides* species showed a high proportion
in the control group. No unique bacterial species were found in either of the two groups.
Seventeen species showed differences between groups, and it was confirmed that the proportions
of *L. acidophilus* and *E. flexneri* were increased in *L. acidophilus* group (Figure 4F).

**Functional profiling and correlation analysis**

Functional profiling was performed at the KEGG level 3 to estimate the effect of the

differential composition of intestinal microbes on the mice (Figure 5). By calculating the LDA
score, it was confirmed that the two groups showed significantly different patterns in 9 categories.
All nine categories were predicted to be more activated in *L. acidophilus* group. The
Phosphotransferase system (PTS) scored the highest, followed by *Staphylococcus aureus*
infection, Synthesis and degradation of ketone bodies.

To further estimate the influence of the altered gut microbiota, Spearman's correlation
analysis of cognitive-behavioral abilities and bacterial OTUs, and fermentation products were
performed (Figure 6). *L. acidophilus* and *E. flexneri* showed a positive correlation with all
assessments of cognitive abilities, while the other 14 OTUs presented a negative correlation. In
particular, step-through latency at Day 2 and Step latency difference for 2 days of the PAT results
showed a significant negative correlation with the *Gemella massiliensis* ($r = -0.8379$, $p =$
0.03248 and $r = -0.8182$, $p = 0.0376$) and *Desulfotomaculum nigrificans* ($r = -0.8781$, $p =$
0.01914 and $r = -0.8450$, $p = 0.03225$).

To provide evidence to indirectly infer the mechanism of action of the gut microbiome, the
concentration of SCFA in the microbial culture was measured (Table S2). Lactic acid and acetic
acid were found in three microbial cultures. Lactic acid was identified in the highest
concentration in *Lcb. paracasei* EG005, and acetic acid was included in the highest concentration
in *L. acidophilus* EG004 culture. Propionate and butyrate were not within detectable ranges.

**Comparative analysis of genetic contents in bacterial whole genome sequences**

To identify its safety and functionality, several genetic factors were detected. Fourteen
genomic islands, two prophage regions, one CRISPR region, and three bacteriocins were found

in the genome of *L. acidophilus* EG004. In *Lcb. paracasei* EG005, 29 genomic islands, 7
prophage regions, 3 CRISPR regions, and 2 bacteriocins were detected. In the case of *Lcb.*
*rhamnosus* EG006, 23 genomic islands, 8 prophage regions, 3 CRISPR regions, and 1
bacteriocin were found in the genome. To estimate a genetic factor related to cognitive ability,
protein annotation was conducted (Figure 7A). Protein metabolism, Carbohydrates, Amino acids
and derivatives showed high proportions, but there was a difference in order by bacterial strains.
Protein metabolism had the highest proportion in *L. acidophilus* EG004 and carbohydrates
presented the highest proportion in *Lcb. paracasei* EG005 and *Lcb. rhamnosus* EG006. In a
subcategory comparison of predicted functional sequences, a difference of genetic contents was
found (Figure 7B). CDSs related to Fatty acids were found in the genomes of *Lcb. paracasei*
EG005 and *Lcb. rhamnosus* EG006. Genes of 3 subcategories (Aromatic amino acids and
derivatives, Alanine, serine, and glycine, and Proline and 4-hydroxyproline) were detected in *Lcb.*
*rhamnosus* EG006, while genes of 3 other categories in Amino Acids in Derivatives were
contained in only *L. acidophilus* EG004.

Discussion

As interest in Gut-Brain Axis has increased, many types of research in this criterion have
been published. However, it is still unclear about the integral mechanism and which strain has a
positive or negative effect. Therefore, we aimed to develop a new strain that has a positive effect
on the host's cognition, and we found 3 strains that caused positive effects in 4 different
cognitive tests (Figure 3). *Lcb. paracasei* group showed improved cognitive ability in the novel
object recognition test. A previous study indicated that this bacterium prevents age-related
cognitive decline and improves cognitive ability (22). Other strain, *Lcb. rhamnosus*, displayed

[revised manuscript text omitted]

antibacterial activity is the essential property of probiotics, such activity of *L. acidophilus*
against harmful and pathogenic bacteria has been reported. In our previous study, we proved that
*L. acidophilus* EG004 is capable of demonstrating the antimicrobial activity (49). Therefore, we
suggest that the antibacterial activity of *L. acidophilus* EG004 was the potential reason for
cognitive ability enhancement.

In functional profiling analysis, we offered explainable factors for the microbial effect on
the brain. Three KEGG categories were related to toxic chemical degradation: Dioxin
degradation, Xylene degradation, and Caprolactam degradation (Figure 5B). Dioxin, a
neurotoxin, can raise autism and neurodegenerative disease (50, 51). Xylene inhibits normal
protein synthesis of neuronal function and induces instability in the neuronal membrane. When it
is inhaled, psychological deficits can be caused (52, 53). These chemicals are noxious to the
brain, so activation of these chemical degradations would have diminished negative effect in *L.*
*acidophilus* group. Besides, two KEGG categories related to the immune system were found.
One of them is *Staphylococcus aureus* infection, which is known to cause brain abscess. Since
there have been many studies demonstrating that *L. acidophilus* has antimicrobial activity against
*S. aureus*, activation of this category is thought to be due to an increase in the amount of *L.*
*acidophilus*. The function of renal cell carcinoma was predicted in the experimental group. As it
involves not only tumor suppressor genes such as VHL, GH, and BHD, but also oncogenes such

as MET and PRCC-TFE3, it seems to be necessary to confirm the exact mechanism and side
effects.

The purpose of this study was to develop a new strain that can improve cognitive ability and
to provide an underlying biological mechanism affecting the brain by the gut microbiome. It is
necessary to measure metabolite changes in order to provide an understanding of the mechanism
of altered cognitive ability. However, altered metabolite from animal body was not fully
identified. To overcome this limitation, we conducted the metagenome analysis, correlation
analysis between cognitive ability and gut microbiome, measurement of SCFA producing ability,
and whole-genome comparison analysis. These analyses were not covered to identification of a
biological factor caused improved cognitive ability, but presented a group of genes and
mechanisms that can infer the process. Although we did not provide direct evidence of phenotype
changes caused by probiotics ingestion, we hope that our findings will help infer the process of
the brain-gut axis.

**Materials and Methods**

**Animals**

4-week-old male C57BL/6 mice (n = 48, average weight 26g) were gained from YoungBio
(Seongnam, Korea). All mice were housed in a group of four per cage under standard controlled
laboratory conditions (temperature of 20±5°C, humidity of 55~60%) on a 12-h light/dark cycle
(light on at 7:00 a.m.). Each group was constituted of 12 mice, and it was nurtured by
distributing 4 mice to 3 cages. Twelve cages were located at random. All animals received *ad*
*libitum* access to food. All animal experiments were performed following protocols approved by
the Institutional Animal Care and Use Committee (IACUC) of Seoul National University, and the
permission number is SNU-190607-4-3.

**Bacterial treatment**

The bacterial strains were isolated from fermented dairy foods. When identifying the brain-
gut axis effect, the important factors to be considered were viability and adherence capacity.
Therefore, we selected the species that are known to have adherence capacity in the GI tract, as
well as the potential for gut-brain axis effect. To identify species of each strain, 16S rRNA genes
were sequenced by Macrogen Inc. (Seoul, Korea) with 27F and 1492R primers. Obtained
sequences were compared with sequences in the NCBI database using BLAST. The experiment
was constituted with 4 groups; 3 experimental groups were fed on autoclaved tap water mixed
with *L. acidophilus* EG004, *Lcb. paracasei* EG005, and *Lcb. rhamnosus* EG006, and a control
group was fed on sterilized tap water. Each group consisted of 12 mice. Bacteria to delivery were

freshly cultivated every day. Probiotic colonies were sub-cultured into 5ml MRS broth for 8
441 hours. After the sub-culture, 3 probiotic strains were inoculated in 500 ml MRS broth for 16
442 hours. Cultivated cells were spun down by centrifugation **with** 4,000 rpm for 10 min. The
443 supernatant was removed, and the pellet was suspended by 0.85 % NaCl solution. Re-suspended
cells were centrifuged **by** 4,000 rpm for 10 min to remove medium ingredients. The washing
process was conducted twice. Washed cells were dissolved into autoclaved tap water. The final
cell concentration of vehicles was about 1.0E9 CFU/ml. To estimate the probiotics amount per
447 day per subject, daily water intake and probiotic concentration in vehicles were recorded. Cell
viability of probiotics was measured by serial dilution and spreading in MRS agar plate. The
probiotics amount per day per subject was calculated as an average of daily water intake per
subject, by multiplying the average of daily probiotic concentration.

**Animal treatment**

[revised manuscript text omitted]

agencourt AMPure XP cleanup (Beckman Coulter, CA, USA), Quant-iT™ PicoGreen™ dsDNA
Assay Kit (Invitrogen, Ireland), and 0.7% agarose gel. The PCR products were diluted and end-
repaired using NEBNext FFPE Repair Mix (New England BioLabs, Ipswich, USA). The
amplicon was Nick-repaired using NEBNext End repair/dA-tailing Module (New England
BioLabs), prior to adapter ligation by NEBNext Quick Ligation Module (New England BioLabs).
The sequencing library was loaded on primed Flongle flow cell according to Nanopore protocol.
Sequencing was performed by MinION MK1b. Sequencing data was acquired by MinKNOW
software (19.12.5) without live base-calling.

**Metagenome analysis**

Raw data were obtained as fast5 files. Base-calling was carried out by Guppy 4.0.11 with
2,000 chunk size and 4 base callers (54). Porechop version 3 was executed for trimming adapter
sequences (<https://github.com/rrwick/Porechop>). To annotate bacterial taxonomy, trimmed
sequences were aligned with reference data from GTDB using Minimap2 (55). In Operational
Taxonomic Unit (OTU) identification, only results with more than 2,500 matching bases and
more than 3,500 bases including gaps in mapping were used. To normalize abundance data, the
TMM (The trimmed mean of M-values) method was used by the edgeR package of R software
(56). To characterize each group, biological diversity was calculated through the physeq package
of R software (57). A rarefaction curve was constructed to check the saturation of genome
sequencing. To compare species richness, alpha diversity was calculated as chao1 and Shannon
indexes. To compare between groups, beta diversity was calculated using Bray-Curtis
dissimilarity and Unifrac distance. P-value was calculated by the Adonis test. For detection of

unequal features, Wilcoxon rank-sum test was performed in each taxonomic level with 0.95
confidence level. To compare functional profile, PICRUSt2 was performed (58). Correlation
between cognitive ability and bacterial OTUs was inferred by Spearman's rank correlation
analysis. P values were adjusted by FDR method.

**SCFA identification in bacterial culture**

To identify the amount of short-chain fatty acids (SCFAs), high-performance liquid
chromatography (HPLC) was performed using Ultimate3000 (Thermo Dionex, USA) and
Aminex 87H column (300x10mm, Bio-Rad, USA). Bacterial cultures of EG004, EG005, and
EG006 were inoculated for 24 hours. After cultivation, the samples were filtered with 0.45 μm of
a membrane filter. The filtered sample of 10 μL was injected into the HPLC.

**Whole-genome sequencing and assembly of EG005 and EG006**

[revised manuscript text omitted]

48. Petrov V, Alifirova V, Saltykova I, Zhukova I, Zhukova N, Dorofeeva YB, Tyakht A, Altukhov I,
Kostryukova E, Titova MJBOSM. 2017. Comparison study of gut microbiota in case of
Parkinson's disease and other neurological disorders. 15:113-125.

49. Kim J, Kim H, Jeon S, Jo J, Kim Y, Kim HJAS. 2020. Synergistic Antibacterial Effects of Probiotic
Lactic Acid Bacteria with Curcuma longa Rhizome Extract as Synbiotic against Cutibacterium
acnes. 10:8955.

50. Ames J, Warner M, Brambilla P, Mocarelli P, Satariano WA, Eskenazi BJE. 2018. Neurocognitive
and physical functioning in the Seveso Women's Health Study. 162:55-62.

51. Guo Z, Xie HQ, Zhang P, Luo Y, Xu T, Liu Y, Fu H, Xu L, Valsami-Jones E, Boksa PJEi. 2018.
Dioxins as potential risk factors for autism spectrum disorder. 121:906-915.

52. Kandyala R, Raghavendra SPC, Rajasekharan STJJoo, JOMFP mp. 2010. Xylene: An overview of
its health hazards and preventive measures. 14:1.

53. Savolainen H, Pfäffli PJAot. 1980. Dose-dependent neurochemical changes during short-term
inhalation exposure to m-xylene. 45:117-122.

54. Wick RR, Judd LM, Holt KEJGb. 2019. Performance of neural network basecalling tools for
Oxford Nanopore sequencing. 20:1-10.

55. Li HJB. 2018. Minimap2: pairwise alignment for nucleotide sequences. 34:3094-3100.
56. Chen Y, McCarthy D, Robinson M, Smyth GKJBUsgAohwboprbeidep. 2014. edgeR: differential
expression analysis of digital gene expression data User's Guide.
57. McMurdie PJ, Holmes SJPo. 2013. phyloseq: an R package for reproducible interactive analysis
and graphics of microbiome census data. 8:e61217.
58. Douglas GM, Maffei VJ, Zaneveld J, Yurgel SN, Brown JR, Taylor CM, Huttenhower C, Langille
MGJB. 2020. PICRUSt2: An improved and customizable approach for metagenome
inference.672295.

**Tables**789 **Table 1. Metagenomic sequencing **statistic** of *L. acidophilus* group and control**

	The number of samples	Total number of reads	Estimated base (Mb)	N50	Total number of counts	Total number of OTUs
LA ^a	5	312,384±31,887	1,434±143	4,872±90	252401.6±25,171	528.4±40
W ^b	5	335,356±45,814	1,485.6±215	4,748±40	259945.6±35,117	539.8±25
Total	10	323,870±37,604	1,459.8±173	4810±72	256173.6±28,860	534.1±32

790 ^a: *L. acidophilus* group, ^b: control group. There was no significant difference between groups. All
791 values were presented as average ± standard error of the mean. Fecal samples compiled after 8
792 weeks of probiotic ingestion were used for metagenome sequencing.

**Figure legends**

**Figure 1. Schematic diagram of the study to discover a new probiotic strain with improved**
 **cognitive ability**

The diagram displays the experimental schedule by day and week for identifying probiotic strain
 with improved cognitive ability. Cognitive ability was measured once a week by four behavioral
 tests. The diagram of each experiment shows the first position of the animal.

**Figure 2. Measurement of additional effect after probiotic consumption**

Experimental groups are expressed in abbreviations. LA: *L. acidophilus* group, LPA: *Lcb.*

*Paracasei* group, LR: *Lcb. Rhamnosus* group, and W: tap water-fed group (control). (A) The

average daily water intake. All groups showed a similar average. (B) The change of daily intaken

probiotic amount by timeline. *L. acidophilus* was ingested in smaller amounts compared to the

other two strains. (C) The average body weight change for 8 weeks. All groups showed similar

averages.

**Figure 3. Results of cognitive behavioral tests**

Experimental groups are expressed in abbreviations. LA: *L. acidophilus* group, LPA: *Lcb.*

*Paracasei* group, LR: *Lcb. rhamnosus* group, and W: the group fed on tap water (control). (A)

Total arm entries during spontaneous alternation test. (B) Spontaneous alternation. This is the

representative value of spontaneous alternation test. (C) Discrimination ratio. It is the

representative value of the novel object recognition test. (D) Comparison of the total time to

observe two objects. (E) Step-through latency of day 1. (F) Step-through latency of day 2. This is

the representative result of the passive avoidance task. (G) Total arm entries during forced

alternation test. (H) Forced alternation. This result is a representative value of forced alternation.

All comparison of average between experimental groups was measured by Wilcoxon rank-sum

test. Significant difference is presented with symbol (Adjusted P-value* < 0.05).

**Figure 4. Results of metagenomics sequencing**

Experimental groups are expressed in abbreviations. LA: *L. acidophilus* group and W: the group

fed on tap water (control). (A) Rarefaction curve of metagenome sequencing. (B) Alpha-diversity

of the *L. acidophilus* group and control. (C) Beta-diversity using Bray-Cutis distance between

the *L. acidophilus* group and control. (D) Beta-diversity using Unifrac distance between both

groups. (E) Comparison of microbial composition at the phylum level. The blue-colored phylum

with the (*) symbol showed a significant difference compared to the two experimental groups. (F)

Comparison of microbial composition at the species level. *L. acidophilus*: *Lactobacillus*

*acidophilus*, E. flexneri: *Escherichia flexneri*, R. hominis: *Roseburia hominis*, A. equolifaciens:
*Adlercreutzia equolifaciens*, S. massiliensis: *Soleaferrea massiliensis*, Lchn. Eligens:
*Lachnospira eligens*, Lch. Bovis_A: *Lachnobacterium bovis_A*, Lc. Phytofermentans:
*Lachnoclostridium phytofermentans*, Bct. Pectinophilus: *Bacteroides_F pectinophilus*, Lc.
Sp900078195: *Lachnoclostridium sp900078195*, Bt. Massiliensis: *Bittarella massiliensis*, G.
massiliensis: *Gemella massiliensis*, St. auricularis: *Staphylococcus auricularis*, Br. Massiliensis:
*Bariatricus massiliensis*, B. sp002556365: *Bacillus_AW sp002556365*, D. nigrificans:
*Desulfotomaculum nigrificans*. All comparisons of average between experimental groups were
measured by independence t-test. Significant difference is presented with symbol (Adjusted P-
value* < 0.05, P-value** < 0.01).

**Figure 5. Results of functional profiling**

Predictive functional profiling of microbiome. All predicted functions have a positive LDA score

for the *L. acidophilus.* group

**Figure 6. Spearman's rank correlation analysis**

Correlation analysis was conducted to detect association among bacterial OTUs, measured
 cognitive abilities, and fermentation products. The color intensity and circle size show the
 strength of the correlation. Red color represents a negative correlation, and blue color is a
 positive correlation. Only circles with adjusted P-value under 0.01 are illustrated in the matrix.
 Results of cognitive ability evaluation were classified by 4 colors: NOR (purple), FA (blue), PAT
 (deep green), and SA (brown). Significant P values indicated by the symbol * (<0.05) and **
 (<0.01).

**Figure 7. Genomic comparison of 3 probiotic strains**

(A) Functional classification of protein coding sequences. All predicted protein sequences were
 classified by categories by SEED system. (B) Subcategories in [Fatty Acids, Lipids, and
 Isoprenoids] and [Amino Acids and Derivatives]. [Fatty Acids, Lipids, and Isoprenoids]
 subcategory showed yellow-green colored head and [Amino Acids and Derivatives] category
 presented light gray colored head.

(A)**(B)****(C)**
(A)**(B)****(C)****(D)****(E)****(F)****(G)****(H)**
(A)**(B)****(C)****(D)****(E)****(F)**
KEGG level1 ■ Environmental Information Processing ■ Human Diseases ■ Metabolism

Category

- Species observed differently
- Novel object recognition test
- Forced alternation
- Passive avoidance task
- Spontaneous alternation
- Fermentation product

s__Lactobacillus acidophilus
s__Escherichia flexneri
s__Roseburia hominis
s__Adlercreutzia equolifaciens
s__Soleaferrea massiliensis
s__Lachnospira eligens
s__Lachnobacterium bovis_A
s__Lachnoclostridium phytofermentans
s__Bacteroides_F pectinophilus
s__Lachnoclostridium sp900078195
s__Bittarella massiliensis
s__Gemella massiliensis
s__Staphylococcus auricularis
s__Bariatricus massiliensis
s__Bacillus_AW sp002556365
s__Desulfotomaculum nigrificans
 Discrimination ratio
 Ratio with familiar object
 Forced alternation
 Novel arm ratio
 Step through latency at Day 2
 Step latency difference during 2 days
 Spontaneous alternation
 Total entered entry
 Lactic acid
 Acetic acid
 Citric acid

s__Lactobacillus acidophilus
s__Escherichia flexneri
s__Roseburia hominis
s__Adlercreutzia equolifaciens
s__Soleaferrea massiliensis
s__Lachnospira eligens
s__Lachnobacterium bovis_A
s__Lachnoclostridium phytofermentans
s__Bacteroides_F pectinophilus
s__Lachnoclostridium sp900078195
s__Bittarella massiliensis
s__Gemella massiliensis
s__Staphylococcus auricularis
s__Bariatricus massiliensis
s__Bacillus_AW sp002556365
s__Desulfotomaculum nigrificans
 Discrimination ratio
 Ratio with familiar object
 Forced alternation
 Novel arm ratio
 Step through latency at Day 2
 Step latency difference during 2 days
 Spontaneous alternation
 Total entered entry
 Lactic acid
 Acetic acid
 Citric acid

(A)**Protein proportion [%]****Subcategory****(B)**

EG004 EG005 EG006

Brief Rebuttal to the remarks of the reviewer3

Comment #1 of the reviewer3: The metagenome sequencing data (16S-23S rRNA) should be submitted to GenBank if it is not submitted yet.

Amendment for comment #1

We thank the reviewer for raising this issue. As the reviewer's comment, we have finished uploading the metagenome sequencing data and whole-genome sequence data of *Lcb. paracasei* EG005 and *Lcb. rhamnosus* EG006. For readers and other researchers' access to data, we added a 'Data availability' session with NCBI accession numbers (*Line 636-640, page 31-32*). Circularized genomes of the three probiotics were added with annotation information in the supplementary information (*Supplementary_data, page 5-7*). We expect that this will give more credit to our research and provide a new application to other researchers.

Comment #2 of the reviewer3: The aim of the paper is to study effect on cognitive ability of *Lactobacillus acidophilus* EG004 in healthy mouse and fecal microbiome analysis using full-length 16S-23S rRNA metagenome sequencing.

In the manuscript, the authors studied a bacterial strain *Lactobacillus acidophilus* EG004 with a positive effect on cognitive ability using a healthy animal model. The authors experimentally verified improved cognitive ability by cognitive behavioral tests. The authors performed full 16S-23S rRNA sequencing and provided gut microbiome composition at a species level. The provided microbiome composition consisted of candidate microbial groups as a biomarker that shows positive effects on cognitive ability. Therefore, their study suggests a new perspective for probiotic strain use applicable for medicine.

The uniqueness of the text is 90% by AntyPlagiarism.net.

The manuscript is written well. English is proper, well understandable.

Reviewer has some comments:

- Line 74 - most of researches were - should be - most of the researches was.
- Line 73 - many researches - should be - many pieces of research.
- Line 82 - industrialization process - should be - industrialization processes.
- Line 105 - for the sentence - Autism, Alzheimer's disease, and Parkinson's disease (7-9) - add additional citation (Danilenko et al., 2021) and add to the References - Danilenko, V.N., Devyatkin, A.V., Marsova, M.V., Shibilova, M.U., Ilyasov, R.A., Shmyrev, V.I., 2021b. Common inflammatory mechanisms in COVID-19 and Parkinson's diseases: the role of microbiome and probiotics in their prevention. *Journal of Inflammation Research* 14, (In press). doi: 10.2147/JIR.S333887.
- Line 108 -to the sentence - the neural pathways of the brain-gut axis (10). - add additional citation (Fetissov et al., 2019). and

add to the References - Fetissov, S.O., Averina, O.V., Danilenko, V.N., 2019. Neuropeptides in the microbiota-brain axis and feeding behavior in autism spectrum disorder. *Nutrition* 61, 43-48. doi: 10.1016/j.nut.2018.10.030.

- Line 112 - Second, the second suggestion - should be - Second, the suggestion
- Line 113 - microbiome affect brain - should be - microbiome affects brain.
- Line 113 - metabolic pathway - should be - metabolic pathways.
- Line 127 - remove one dot.
- Line 153 - The averages daily - should be - The averages of daily.
- Line 168 - In the comparison of - should be - The comparison of.
- Line 194 - light room - should be - lightroom.
- Line 195 - remove italics of the word - group.
- Line 226 - comparison - should be - comparative.
- Line 236 - familiae - should be - families.
- Line 275 - whole genome - should be - whole-genome.
- Line 308 - recognition test and passive avoidance task - should be - recognition tests and passive avoidance tasks.
- Line 321 - were - should be - was.
- Line 343 - factor - should be - factors.
- Line 350 - purpose - should be - purposes.
- Line 370 - these evidences - should be - this evidence.
- Line 398 - negative effect - should be - negative effects.
- Line 408 - to provide - should be - provide.
- Line 413 - These analyses were not covered to identification of a biological factor caused - should be - These analyses were not covered in the identification of a biological factor that caused.
- Line 416 - probiotics ingestion - should be - probiotic ingestion.
- Line 444 - by - should be - at.
- Line 442 - with - should be - at.
- Line 453 - add space after dot.
- Line 457 - from probiotics intake - should be - after probiotic intake.
- Line 459 - room condition - should be - room conditions.
- Line 472 - rodent's habit - should be - rodents' habits.
- Line 478 - entered - should be - that entered.
- Line 486 - preference - should be - preferences.
- Line 516 - After 1 minute for adaptation - should be - After 1 minute of adaptation.
- Line 531 - time taken - should be - time is taken.
- Line 554 - correction - should be - corrections.
- Line 789 - statistic - should be - statistics.

Please check English by professional translator one more times.

In further authors should study details of biological factors and molecular mechanisms that caused improved cognitive ability in mice after treatment with *L. acidophilus* EG004 strain.

Amendment for comment #2

Thank you for reading carefully and giving us kind advice. This is the kindest comment we've ever received. Based on the reviewer's comments, we revised the manuscript. However, the paper the reviewer recommended was not found because the paper was not published yet. So, we added another paper that indicated the relationship between the gut microbiome and Parkinson's disease (Danilenko VN, Stavrovskaya AV, Voronkov DN, Gushchina AS, Marsova MV, Yamshchikova NG, Ol'shansky AS, Ivanov M, Ivanov M, Illarioshkin SNJAoC, Neurology E. 2020. The use of a pharmabiotic based on the *Lactobacillus fermentum* U-21 strain to modulate the neurodegenerative process in an experimental model of Parkinson disease). We expect it to help the readers understand our contents. To deliver accurately, grammatical errors have been corrected throughout the entire manuscript again. As the reviewer mentioned, we are designing a further

**Dept. of Agricultural Biotechnology and Research
Institute of Agriculture and Life Sciences
Seoul National University
San 56-1 Daehak-dong, Gwanak-gu, Seoul 151-742
Republic of Korea**

study including measurement of metabolite changes to understand the biological mechanism accurately. We hope to report positive results again in the near future.

November 29, 2021

Prof. Heebal Kim
Seoul National University
Seoul
Korea (South), Republic of

Re: Spectrum01815-21R1 (Positive effect on cognitive ability of *Lactobacillus acidophilus* EG004 in healthy mouse and fecal microbiome analysis using full-length 16S-23S rRNA metagenome sequencing)

Dear Prof. Heebal Kim:

Thank you for submitting your manuscript to Microbiology Spectrum. As you will see your paper is very close to acceptance. Please modify the manuscript along the lines recommended by Reviewer 3 (see below). As these revisions are quite minor, I expect that you should be able to turn in the revised paper in less than 30 days, if not sooner. If your manuscript was reviewed, you will find the reviewers' comments below.

When submitting the revised version of your paper, please provide (1) point-by-point responses to the issues I raised in your cover letter, and (2) a PDF file that indicates the changes from the original submission (by highlighting or underlining the changes) as file type "Marked Up Manuscript - For Review Only". Please use this link to submit your revised manuscript. Detailed instructions on submitting your revised paper are below.

Link Not Available

Sincerely,

Jan Claesen

Reviewer comments:

Reviewer #3 (Comments for the Author):

Reviewer comments

Manuscript: Spectrum01815-21R1 Positive effect on cognitive ability of *Lactobacillus acidophilus* EG004 in healthy mouse and fecal microbiome analysis using full-length 16S-23S rRNA metagenome sequencing

The manuscript was improved but I have some questions:

Line 54 - Cognition is one of the functions of the brain. The authors should write in the Manuscript the idea that they study bacterial strain that has positive effects on brain function, which can be recognized through changes in cognitive processes.
Line 68 - In the annotation, you do not say a word about strains EG005 and EG006. Why? Also, add into the discussion part more information about comparison and differences in the action of these three strains. Explain the reasons for these differences.

Line 130 - will be better if you use the word - healing effects

Line 150 - what kind of molecular method? add the explanation into the text.

Line 390 - you wrote - that the antibacterial activity of *L. acidophilus* EG004 was the potential reason for cognitive ability enhancement. - how it is possible? Why do you assume this?

Line 407 - Line 54 - Cognition is one of the functions of the brain. The authors should write in the Manuscript the idea that they study bacterial strain that has positive effects on brain function, which can be recognized through changes in cognitive processes.

Line 421 - why male?

Line 605 - Why is EG004 do not present here?

Line 623 - Add here the information from Data availability - The complete sequences of *Lcb. paracasei* EG005 and *Lcb. rhamnosus* EG006 is available in the NCBI database with accession numbers, SAMN23227569 and SAMN23227570, respectively. The metagenomic sequences are available in the NCBI database under the accession number PRJNA781018. Please answer my question and add information to the Manuscript.

I added the PDF file with highlighted comments.

No other comments.

A minor revision is required.

Preparing Revision Guidelines

- point-by-point responses to the issues I raised in your cover letter
- Upload a compare copy of the manuscript (without figures) as a "Marked-Up Manuscript" file.
- Each figure must be uploaded as a separate file, and any multipanel figures must be assembled into one file.
- Manuscript: A .DOC version of the revised manuscript
- Figures: Editable, high-resolution, individual figure files are required at revision, TIFF or EPS files are preferred

Please return the manuscript within 60 days; if you cannot complete the modification within this time period, please contact me. If you do not wish to modify the manuscript and prefer to submit it to another journal, please notify me of your decision immediately so that the manuscript may be formally withdrawn from consideration by Microbiology Spectrum.

**Title**

Positive effect on cognitive ability of *Lactobacillus acidophilus* EG004 in healthy
mouse and fecal microbiome analysis using full-length 16S-23S rRNA
metagenome sequencing

**Running title**

Positive effect on cognitive ability of *L. acidophilus*

**Authors**

Soomin Jeon^{a,†}, Hyaekang Kim^{a,†}, Jina Kim^a, Donghyeok Seol^{a,b}, Jinchul Jo^a,
Youngseok Choi^a, Seoae Cho^b, Heebal Kim^{a,b,d,*}

11 ^aDepartment of Agricultural Biotechnology and Research Institute of Agriculture
and Life Sciences, Seoul National University, Seoul, Republic of Korea

13 ^beGnome, Inc, Seoul, Republic of Korea

14 ^dInterdisciplinary Program in Bioinformatics, Seoul National University, Seoul,
Republic of Korea

*Corresponding authors

† Soomin Jeon and Hyaekang Kim contributed equally to this work. Author order

was determined retroalphabetically.

Soomin Jeon

Email: soty23@snu.ac.kr

Hyaekang Kim

Email:hkim458@snu.ac.kr

Jina Kim

Email:jinak750@gmail.com

Donghyeok Seol

Email:sdh1621@snu.ac.kr

Jinchul Jo

Email: macjoo2000@snu.ac.kr

Youngseok Choi

Email:seok1213neo@snu.ac.kr

Seoae Cho

Email:seoae@egenome.co.kr

Heebal Kim

Email: heebal@snu.ac.kr

*Correspondence

Heebal Kim, Department of Agricultural Biotechnology and Research Institute of

Agriculture and Life Sciences, Seoul National University, Seoul, Republic of

Korea

Email: heebal@snu.ac.kr

Tel.: +82-2-880-4822

Fax: +82-2-883-8812

**Word count**

Abstract; 249 words

Text; 4,245 words (excluding materials and methods) and 6,952 words (including

materials and methods)

**Abstract**

The concept of the 'Gut-brain axis' has risen. Many types of research demonstrated the effect and
mechanism of the GBA. Although many studies have been reported, most of the studies are
focused on neurodegenerative disease and it is still not clear what type of bacterial strains have
positive effects on the brain. Therefore, we designed an experiment to discover a strain that
positively affects cognitive ability using healthy mice. The experimental group consisted of a
control group and three probiotic consumption groups, *Lactobacillus acidophilus*,
*Lacticaseibacillus paracasei*, and *Lacticaseibacillus rhamnosus*, which are verified to have
beneficial effects for host health as the gut microbiome. Cognitive ability was measured by 4
cognitive-behavioral tests and the group fed on *L. acidophilus* showed the most improved
cognitive ability. To provide an understanding of the altered microbial composition effect on the
brain, we performed full 16S-23S rRNA sequencing using Nanopore, and OTUs were identified
at a species level. In the group fed on *L. acidophilus*, the intestinal bacterial ratio of Firmicutes
and Proteobacteria phyla increased and the bacterial proportions of 16 species were significantly
different from those of the control group. We estimated that the positive results on the cognitive
behavioral tests were due to the increased proportion of *L. acidophilus* EG004 strain in the
subjects' intestines since the strain is capable of producing butyrate and therefore modulating
neurotransmitters and neurotrophic factors. We expect that our new strain expands the industrial
field of *L. acidophilus* and helps understand the mechanism of the brain-gut axis.

**Importance**

In recent, the concept of 'gut-brain axis' has risen that microbes in the GI tract affect brain by

modulating signal molecules. Although many pieces of research were reported in a short period,
a signaling mechanism and effect of a specific bacterial strain are still unclear. Besides, since
most of the researches was focused on neurodegenerative disease, the study with a healthy
animal model is still insufficient. In this study, we provide a bacterial strain (*Lactobacillus*
*acidophilus* EG004) with a positive effect on cognitive ability using a healthy animal model. We
experimentally verified improved cognitive ability by cognitive behavioral tests. We performed
full 16S-23S rRNA sequencing using Nanopore MinION, and provided gut microbiome
composition at a species level. The provided microbiome composition consisted of candidate
microbial groups as a biomarker that shows positive effects on cognitive ability. Therefore, our
study suggests a new perspective for probiotic strain use applicable for various industrialization
processes.

**Keywords**

*Lactobacillus acidophilus*, gut microbiome, gut-brain axis, cognitive ability, Nanopore
sequencing

**Introduction**

The human body is a complex community that habituates various bacteria. Among the
bacterial communities in the human body, the gastrointestinal tract is the best bacterial
community that has the most abundant and various bacteria (1). In 2006, having been released
research that obesity is associated with bacterial composition in the gut, a study for gut
microbiome began in earnest (2). The gut microbiome is defined as the collective genomes of
microorganisms that live in the gastrointestinal tract. Functions of the gut microbiome have been
reported such as nutrient metabolism and regulation of the immune system for the host (3).
Microbial composition in the gut is altered by environmental factors like age, diet, stress, and
lifestyle, and the change in microbial composition can induce physical changes in the host (4). In
recent, the gut microbiome's effects on the brain have been proved and the concept of the brain-
gut axis has risen to the surface (5). The brain-gut axis is a complex system involving the enteric
nervous system and central nervous system including the brain and spinal cord, and it works with
bidirectional communication between the central and the enteric nervous system (6). Although
the brain is located apart from the gut, the gut microbiome can affect the brain by stimulating the
enteric nervous system and vagus nerve. Thus, dysbiosis of the gut microbiome often causes
brain diseases. The recent experimental results described that gut microbiome dysbiosis was
observed in patients with Autism, Alzheimer's disease, and Parkinson's disease (7-10). At the
same time, studies on the mechanisms to understand the brain-gut axis have been conducted.
First, it was suggested that the microbial-derived metabolites are the main components acting on
the neural pathways of the brain-gut axis (11, 12). The most well-studied substances are short-
chain fatty acids (SCFA) such as acetate, propionate, and butyrate, which are produced in the
process of decomposing non-digestible fibers and carbohydrates (13). It promotes indirect

signaling to the brain by modulation and induction of neurotransmitter and neurotrophic factors
like γ -aminobutyric acid (GABA) and Brain-derived neurotrophic factor (BDNF). Second, the
suggestion was that the gut microbiome affects brain function by regulating metabolic pathways
(14). Previous research reported that the level of tryptophan metabolites including serotonin and
indolepyruvate was altered by the gut microbiome. These metabolites have roles in the
functioning of the gut-brain axis such as signaling and anti-oxidant. Third, the gut microbiome
may affect the brain by immune pathway (15). Interferon (IFN), Tumor necrosis factor (TNF),
and Interleukin are well-known immune factors. According to recent studies, the amount of the
immune factors is regulated by the intestinal microflora. These immune factors affect brain
function by stimulating and activating the hypothalamic-pituitary-adrenal axis. Finally, it was
suggested that gut microbes directly influence the brain by altering the fatty acid composition of
the brain (16). Several studies have been reported on the correlation between intestinal
microbes and the brain, but the specific mechanism of the brain-gut axis is still not clear.

Probiotics are defined as bacteria that have positive effects on the host body (17). Probiotics
have been widely used as a health supplement since it has various beneficial functions to host's
health with high adhesion property to the intestine and low side effect. Most probiotics include
bacteria genera that are gram-positive, facultative anaerobic and rod-shaped. *Lacticaseibacillus*
*rhamnosus* (*Lcb. rhamnosus*) is one of the longest-studied probiotic species, and many strains
such as LGG and GR-1 belonging to this genus are commercially available. It is well known that
*Lcb. rhamnosus* has positive effects on diarrhea, acute gastroenteritis, and atopic dermatitis (18-
20). Recently, its neurobehavioral effects such as anxiety and depression relief have been
reported (21). *Lacticaseibacillus paracasei* (*Lcb. paracasei*) is one of the representative probiotic
species, and it has been studied to be effective in treating ulcerative colitis and allergic rhinitis

(22, 23). In a recent study, an effect on age-related cognitive decline and a stress relief effect was
reported with several strains of this species (24). *Lactobacillus acidophilus* (*L. acidophilus*) is
another representative probiotic strain. This strain lowers cholesterol levels and has beneficial
health effects such as antibacterial effects against harmful bacteria like *Streptococcus mutans* and
*Salmonella typhimurium* (25, 26).

In this study, we aimed to present a new strain that has an enhancing effect on cognitive
ability through the brain-gut axis and provide an additional understanding of the brain-gut axis.
Three probiotic strains, *L. acidophilus*, *Lcb. paracasei*, and *Lcb. rhamnosus*, which have
previously demonstrated beneficial effects to the host as one of the gut-microbiome strains, were
used to confirm their positive effects on cognitive ability. Full 16S and 23S rRNA sequencing
was performed to annotate the gut microbiome at a species level for downstream analysis. We
expect our results to provide an understanding of the role of the gut microbiome.

**Results**

**Bacterial and animal treatments**

Three probiotic strains, *L. acidophilus* EG004, *Lcb. paracasei* EG005, and *Lcb. rhamnosus*
EG006, have been identified by the molecular method. These strains were clustered with
available *L. acidophilus*, *Lcb. paracasei*, and *Lcb. rhamnosus* strains, respectively, in a
phylogenetic tree of 16S rRNA gene (Figure S1). Probiotic strains were consumed by mice for 8
153 weeks with assessments of cognitive ability (Figure 1). The averages of daily water intake per
154 subject were similar between groups (Figure 2A). Daily probiotic intakes were maintained
constantly and the average amount of *L. acidophilus* group, *Lcb. paracasei* group, and *Lcb.*
*rhamnosus* group were calculated as $(7.82E09 \pm 1.95E09)$, $(4.37E10 \pm 5.17E09)$, and
$(3.74E10 \pm 3.98E09)$ CFUs (Figure 2B). To identify the additional effect of probiotics, the body
weights of mice were measured every week (Figure 2C and S2). Patterns of weight gain in the 4
groups were similar for 8 weeks. The mean body weight gains of the control group showed the
highest value, which was 9.08 g. *Lcb. paracasei* group showed a significant difference from the
control group with P-value under 0.05 in the second measurement, but the difference was
immediately recovered. Similar to weekly weight change, statistical significance was not found
in accumulated weight between experimental groups for 8 weeks.

**Cognitive behavioral tests**

Spontaneous alternation test was conducted to assess spatial learning and short-term
memory. Although the average number of the total entries to each arm in *Lcb. paracasei* group

was slightly low, the difference between groups was not found (Figure 3A). The comparison of
the mouse ratio showed spontaneous alternation for the first 3 entries, *L. acidophilus* group
showed the highest value as 75.0%. (Table S1). In spontaneous alternation, the average values of
probiotics-fed groups were higher than the value of the vehicle-fed group (Figure 3B). Among
the 4 experimental groups, *L. acidophilus* group showed the highest alternation ratio. Wilcoxon
rank-sum test was performed to identify statistical significance, but there was no statistical
difference between the experimental groups and control group.

Novel object recognition (NOR) test was performed to evaluate long-term and explicit
memory using 4 different features (Figure 3C, 3D, and Table S1). *L. acidophilus* group exhibited
the highest average ratio of mouse that touched the novel object before the familiar object,
whereas *Lcb. rhamnosus* group showed the lowest value under the control group. At
discrimination ratio comparison, the three probiotics-fed groups showed higher average values
than the control, and *L. acidophilus* group showed the highest values. To identify if there is a
significant difference, Wilcoxon rank-sum test was performed. When compared to the vehicle-
fed group, *L. acidophilus* and *Lcb. paracasei* groups displayed statistically significant differences

[revised manuscript text omitted]

antibacterial activity is the essential property of probiotics, such activity of *L. acidophilus* against
harmful and pathogenic bacteria has been reported. In our previous study, we proved that *L.*
*acidophilus* EG004 is capable of demonstrating the antimicrobial activity (51). Therefore, we
suggest that the antibacterial activity of *L. acidophilus* EG004 was the potential reason for
cognitive ability enhancement.

In functional profiling analysis, we offered explainable factors for the microbial effect on
the brain. Three KEGG categories were related to toxic chemical degradation: Dioxin
degradation, Xylene degradation, and Caprolactam degradation (Figure 5B). Dioxin, a
neurotoxin, can raise autism and neurodegenerative disease (52, 53). Xylene inhibits normal
protein synthesis of neuronal function and induces instability in the neuronal membrane. When it
is inhaled, psychological deficits can be caused (54, 55). These chemicals are noxious to the
brain, so activation of these chemical degradations would have diminished negative effects in *L.*
*acidophilus* group. Besides, two KEGG categories related to the immune system were found.
One of them is *Staphylococcus aureus* infection, which is known to cause brain abscess. Since
there have been many studies demonstrating that *L. acidophilus* has antimicrobial activity against
*S. aureus*, activation of this category is thought to be due to an increase in the amount of *L.*
*acidophilus*. The function of renal cell carcinoma was predicted in the experimental group. As it
involves not only tumor suppressor genes such as VHL, GH, and BHD, but also oncogenes such

as MET and PRCC-TFE3, it seems to be necessary to confirm the exact mechanism and side
effects.

The purpose of this study was to develop a new strain that can improve cognitive ability and
provide an underlying biological mechanism affecting the brain by the gut microbiome. It is
necessary to measure metabolite changes in order to provide an understanding of the mechanism
of altered cognitive ability. However, altered metabolite from animal body was not fully
identified. To overcome this limitation, we conducted the metagenome analysis, correlation
analysis between cognitive ability and gut microbiome, measurement of SCFA producing ability,
and whole-genome comparison analysis. These analyses were not covered in the identification of
a biological factor that caused improved cognitive ability, but presented a group of genes and
mechanisms that can infer the process. Although we did not provide direct evidence of phenotype
changes caused by probiotic ingestion, we hope that our findings will help infer the process of
the brain-gut axis.

**Materials and Methods**

**Animals**

4-week-old male C57BL/6 mice (n = 48, average weight 26g) were gained from YoungBio
(Seongnam, Korea). All mice were housed in a group of four per cage under standard controlled
laboratory conditions (temperature of 20±5°C, humidity of 55~60%) on a 12-h light/dark cycle
(light on at 7:00 a.m.). Each group was constituted of 12 mice, and it was nurtured by
distributing 4 mice to 3 cages. Twelve cages were located at random. All animals received *ad*
*libitum* access to food. All animal experiments were performed following protocols approved by
the Institutional Animal Care and Use Committee (IACUC) of Seoul National University, and the
permission number is SNU-190607-4-3.

**Bacterial treatment**

The bacterial strains were isolated from fermented dairy foods. When identifying the brain-
gut axis effect, the important factors to be considered were viability and adherence capacity.
Therefore, we selected the species that are known to have adherence capacity in the GI tract, as
well as the potential for gut-brain axis effect. To identify species of each strain, 16S rRNA genes
were sequenced by Macrogen Inc. (Seoul, Korea) with 27F and 1492R primers. Obtained
sequences were compared with sequences in the NCBI database using BLAST. The experiment
was constituted with 4 groups; 3 experimental groups were fed on autoclaved tap water mixed
with *L. acidophilus* EG004, *Lcb. paracasei* EG005, and *Lcb. rhamnosus* EG006, and a control
group was fed on sterilized tap water. Each group consisted of 12 mice. Bacteria to delivery were

freshly cultivated every day. Probiotic colonies were sub-cultured into 5ml MRS broth for 8
441 hours. After the sub-culture, 3 probiotic strains were inoculated in 500 ml MRS broth for 16
442 hours. Cultivated cells were spun down by centrifugation at 4,000 rpm for 10 min. The
443 supernatant was removed, and the pellet was suspended by 0.85 % NaCl solution. Re-suspended
cells were centrifuged at 4,000 rpm for 10 min to remove medium ingredients. The washing
process was conducted twice. Washed cells were dissolved into autoclaved tap water. The final
cell concentration of vehicles was about 1.0×10^9 CFU/ml. To estimate the probiotics amount per
447 day per subject, daily water intake and probiotic concentration in vehicles were recorded. Cell
viability of probiotics was measured by serial dilution and spreading in MRS agar plate. The
probiotics amount per day per subject was calculated as an average of daily water intake per
subject, by multiplying the average of daily probiotic concentration.

**Animal treatment**

[revised manuscript text omitted]

agencourt AMPure XP cleanup (Beckman Coulter, CA, USA), Quant-iT™ PicoGreen™ dsDNA
Assay Kit (Invitrogen, Ireland), and 0.7% agarose gel. The PCR products were diluted and end-
repaired using NEBNext FFPE Repair Mix (New England BioLabs, Ipswich, USA). The
amplicon was Nick-repaired using NEBNext End repair/dA-tailing Module (New England
BioLabs), prior to adapter ligation by NEBNext Quick Ligation Module (New England BioLabs).
The sequencing library was loaded on primed Flongle flow cell according to Nanopore protocol.
Sequencing was performed by MinION MK1b. Sequencing data was acquired by MinKNOW
software (19.12.5) without live base-calling.

**Metagenome analysis**

Raw data were obtained as fast5 files. Base-calling was carried out by Guppy 4.0.11 with
2,000 chunk size and 4 base callers (56). Porechop version 3 was executed for trimming adapter
sequences (<https://github.com/rrwick/Porechop>). To annotate bacterial taxonomy, trimmed
sequences were aligned with MIRROR (<http://mirror.egnome.co.kr/>) using Minimap2 (57). In
Operational Taxonomic Unit (OTU) identification, only results with more than 2,500 matching
bases and more than 3,500 bases including gaps in mapping were used. To normalize abundance
data, the TMM (The trimmed mean of M-values) method was used by the edgeR package of R
software (58). To characterize each group, biological diversity was calculated through the physeq
package of R software (59). A rarefaction curve was constructed to check the saturation of
genome sequencing. To compare species richness, alpha diversity was calculated as chao1 and
Shannon indexes. To compare between groups, beta diversity was calculated using Bray-Curtis
dissimilarity and Unifrac distance. P-value was calculated by the Adonis test. For detection of

unequal features, Wilcoxon rank-sum test was performed in each taxonomic level with 0.95
confidence level. To compare functional profile, PICRUSt2 was performed (60). Correlation
between cognitive ability and bacterial OTUs was inferred by Spearman's rank correlation
analysis. P values were adjusted by FDR method.

**SCFA identification in bacterial culture**

To identify the amount of short-chain fatty acids (SCFAs), high-performance liquid
chromatography (HPLC) was performed using Ultimate3000 (Thermo Dionex, USA) and
Aminex 87H column (300x10mm, Bio-Rad, USA). Bacterial cultures of EG004, EG005, and
EG006 were inoculated for 24 hours. After cultivation, the samples were filtered with 0.45 μm of
a membrane filter. The filtered sample of 10 μL was injected into the HPLC.

**Whole-genome sequencing and assembly of EG005 and EG006**

[revised manuscript text omitted]

- 50. Petrov V, Alifirova V, Saltykova I, Zhukova I, Zhukova N, Dorofeeva YB, Tyakht A, Altukhov I,
Kostryukova E, Titova MJBOSM. 2017. Comparison study of gut microbiota in case of
Parkinson's disease and other neurological disorders. 15:113-125.
- 51. Kim J, Kim H, Jeon S, Jo J, Kim Y, Kim HJAS. 2020. Synergistic Antibacterial Effects of Probiotic
Lactic Acid Bacteria with Curcuma longa Rhizome Extract as Synbiotic against Cutibacterium
acnes. 10:8955.
- 52. Ames J, Warner M, Brambilla P, Mocarelli P, Satariano WA, Eskenazi BJE. 2018. Neurocognitive
and physical functioning in the Seveso Women's Health Study. 162:55-62.
- 53. Guo Z, Xie HQ, Zhang P, Luo Y, Xu T, Liu Y, Fu H, Xu L, Valsami-Jones E, Boksa PJEi. 2018.
Dioxins as potential risk factors for autism spectrum disorder. 121:906-915.

- 54. Kandyala R, Raghavendra SPC, Rajasekharan STJ, JomFP mp. 2010. Xylene: An overview of
its health hazards and preventive measures. 14:1.
- 55. Savolainen H, Pfäffli PJAot. 1980. Dose-dependent neurochemical changes during short-term
inhalation exposure to m-xylene. 45:117-122.
- 56. Wick RR, Judd LM, Holt KEJGb. 2019. Performance of neural network basecalling tools for
Oxford Nanopore sequencing. 20:1-10.
- 57. Li HJB. 2018. Minimap2: pairwise alignment for nucleotide sequences. 34:3094-3100.
- 58. Chen Y, McCarthy D, Robinson M, Smyth GKJBUsgAohwboprbeidep. 2014. edgeR: differential
expression analysis of digital gene expression data User's Guide.
- 59. McMurdie PJ, Holmes SJPo. 2013. phyloseq: an R package for reproducible interactive analysis
and graphics of microbiome census data. 8:e61217.
- 60. Douglas GM, Maffei VJ, Zaneveld J, Yurgel SN, Brown JR, Taylor CM, Huttenhower C, Langille
MGJB. 2020. PICRUSt2: An improved and customizable approach for metagenome
inference. 672295.
- 61. Grant JR, Stothard PJNar. 2008. The CGView Server: a comparative genomics tool for circular
genomes. 36:W181-W184.

**Tables**804 **Table 1. Metagenomic sequencing statistics of *L. acidophilus* group and control**

	The number of samples	Total number of reads	Estimated base (Mb)	N50	Total number of counts	Total number of OTUs
LA ^a	5	312,384±31,887	1,434±143	4,872±90	252401.6±25,171	528.4±40
W ^b	5	335,356±45,814	1,485.6±215	4,748±40	259945.6±35,117	539.8±25
Total	10	323,870±37,604	1,459.8±173	4810±72	256173.6±28,860	534.1±32

805 ^a: *L. acidophilus* group, ^b: control group. There was no significant difference between groups. All
806 values were presented as average ± standard error of the mean. Fecal samples compiled after 8
807 weeks of probiotic ingestion were used for metagenome sequencing.

**Figure legends**

**Figure 1. Schematic diagram of the study to discover a new probiotic strain with improved**
 **cognitive ability**

The diagram displays the experimental schedule by day and week for identifying probiotic strain
 with improved cognitive ability. Cognitive ability was measured once a week by four behavioral
 tests. The diagram of each experiment shows the first position of the animal.

**Figure 2. Measurement of additional effect after probiotic consumption**

Experimental groups are expressed in abbreviations. LA: *L. acidophilus* group, LPA: *Lcb.*
 *Paracasei* group, LR: *Lcb. Rhamnosus* group, and W: tap water-fed group (control). (A) The
 average daily water intake. All groups showed a similar average. (B) The change of daily intaken
 probiotic amount by timeline. *L. acidophilus* was ingested in smaller amounts compared to the
 other two strains. (C) The average body weight change for 8 weeks. All groups showed similar
 averages.

**Figure 3. Results of cognitive behavioral tests**

Experimental groups are expressed in abbreviations. LA: *L. acidophilus* group, LPA: *Lcb.*
 *Paracasei* group, LR: *Lcb. rhamnosus* group, and W: the group fed on tap water (control). (A)
 Total arm entries during spontaneous alternation test. (B) Spontaneous alternation. This is the
 representative value of spontaneous alternation test. (C) Discrimination ratio. It is the
 representative value of the novel object recognition test. (D) Comparison of the total time to
 observe two objects. (E) Step-through latency of day 1. (F) Step-through latency of day 2. This is
 the representative result of the passive avoidance task. (G) Total arm entries during forced
 alternation test. (H) Forced alternation. This result is a representative value of forced alternation.
 All comparison of average between experimental groups was measured by Wilcoxon rank-sum
 test. Significant difference is presented with symbol (Adjusted P-value* < 0.05).

**Figure 4. Results of metagenomics sequencing**

Experimental groups are expressed in abbreviations. LA: *L. acidophilus* group and W: the group

fed on tap water (control). **(A)** Rarefaction curve of metagenome sequencing. **(B)** Alpha-diversity

of the *L. acidophilus* group and control. **(C)** Beta-diversity using Bray-Cutis distance between

the *L. acidophilus* group and control. **(D)** Beta-diversity using Unifrac distance between both

groups. **(E)** Comparison of microbial composition at the phylum level. The blue-colored phylum

with the (*) symbol showed a significant difference compared to the two experimental groups. **(F)**

Comparison of microbial composition at the species level. *L. acidophilus*: *Lactobacillus*

*acidophilus*, E. flexneri: *Escherichia flexneri*, R. hominis: *Roseburia hominis*, A. equolifaciens:
*Adlercreutzia equolifaciens*, S. massiliensis: *Soleiferrea massiliensis*, Lchn. Eligens:
*Lachnospira eligens*, Lch. Bovis_A: *Lachnobacterium bovis_A*, Lc. Phytofermentans:
*Lachnoclostridium phytofermentans*, Bct. Pectinophilus: *Bacteroides_F pectinophilus*, Lc.
Sp900078195: *Lachnoclostridium sp900078195*, Bt. Massiliensis: *Bittarella massiliensis*, G.
massiliensis: *Gemella massiliensis*, St. auricularis: *Staphylococcus auricularis*, Br. Massiliensis:
*Bariatricus massiliensis*, B. sp002556365: *Bacillus_AW sp002556365*, D. nigrificans:
*Desulfotomaculum nigrificans*. All comparisons of average between experimental groups were
measured by independence t-test. Significant difference is presented with symbol (Adjusted P-
value* < 0.05, P-value** < 0.01).

**Figure 5. Results of functional profiling**

Predictive functional profiling of microbiome. All predicted functions have a positive LDA score

for the *L. acidophilus* group.

**Figure 6. Spearman's rank correlation analysis**

Correlation analysis was conducted to detect association among bacterial OTUs, measured
 cognitive abilities, and fermentation products. The color intensity and circle size show the
 strength of the correlation. Red color represents a negative correlation, and blue color is a
 positive correlation. Only circles with adjusted P-value under 0.01 are illustrated in the matrix.
 Results of cognitive ability evaluation were classified by 4 colors: NOR (purple), FA (blue), PAT
 (deep green), and SA (brown). Significant P values indicated by the symbol * ($P < 0.05$) and **
 ($P < 0.01$).

**Figure 7. Genomic comparison of 3 probiotic strains**

(A) Functional classification of protein coding sequences. All predicted protein sequences were
 classified by categories by SEED system. (B) Subcategories in [Fatty Acids, Lipids, and
 Isoprenoids] and [Amino Acids and Derivatives]. [Fatty Acids, Lipids, and Isoprenoids]
 subcategory showed yellow-green colored head and [Amino Acids and Derivatives] category
 presented light gray colored head.

(A)**(B)****(C)**

KEGG level1 ■ Environmental Information Processing ■ Human Diseases ■ Metabolism

Category

(A)**(B)**

Brief Rebuttal to the remarks of the reviewer3

Comment #1 of the reviewer3: Line 54 - Cognition is one of the functions of the brain. The authors should write in the Manuscript the idea that they study bacterial strain that has positive effects on brain function, which can be recognized through changes in cognitive processes.

Amendment for comment #1

We appreciate the reviewer for pointing out the most important part of understanding the experimental design. The context has been added in the Abstract and Introduction parts, and it will help readers naturally understand the research aim (*Line 54-56, page 4*).

Comment #2 of the reviewer3: Line 68 - In the annotation, you do not say a word about strains EG005 and EG006. Why? Also, add into the discussion part more information about comparison and differences in the action of these three strains. Explain the reasons for these differences.

Amendment for comment #2

Thank you for your valuable advice. In order to focus on *L. acidophilus* EG004, the results of the other two strains were omitted in the abstract of the previous manuscript. To increase the overall understanding of the study, we have added results for *Lcb. paracasei* EG005 and *Lcb. rhamnosus* EG006 to Abstract (*Line 58-60, page 4*). In addition, referring to the reviewer's advice, discussion was revised to provide comparative information on the effects after ingestion of the three strains (*Line 294-319, page 16-17, and Supplementary Table3*). This additional explanation will provide the reader with a richer understanding of the cognitive abilities of each Lactic acid bacteria.

Comment #3 of the reviewer3: Line 130 - will be better if you use the word - healing effects

Amendment for comment #3

Based on the reviewer comments, the word was modified to a more appropriate word (*Line 129-131, page 7*).

Comment #4 of the reviewer3: Line 150 - what kind of molecular method? add the explanation into the text.

Amendment for comment #4

Thanks for your kind comments. The previous expression as “molecular method” has been replaced by “16S rRNA sequencing” (*Line 149-150, page 9*). A detailed description of this method can be found in the Materials and methods section. We expect that this clear statement will help the reader's understanding.

Comment #5 of the reviewer3: Line 390 - you wrote - that the antibacterial activity of *L. acidophilus* EG004 was the potential reason for cognitive ability enhancement. - how it is possible? Why do you assume this?

Amendment for comment #5

We thank the reviewer for raising this issue. We assumed that the low levels of the microorganisms (such as *Adlercreutzia equolifaciens* and *Roseburia hominis*) were affected by ingested *L. acidophilus* EG004. The only difference was the intake of *L. acidophilus* between the control group and the *L. acidophilus* group. Based on the function of *L. acidophilus* indicated in previous studies, we estimated that *L. acidophilus* interfered with the habitat and growth of other microorganisms through preoccupation of habitat and antibacterial activity. However, we did not provide experimental evidence for the process in this study. We acknowledge that the current argument has some leaps and bounds. This may be misleading to readers. Accordingly, we omitted the detailed explanation of the presumed mechanism, leaving only the assumption that *L. acidophilus* may have been affected with toning down of suggestion (*Line 391-395, page 20*). The revised manuscript will be able to more accurately convey the effects of *L. acidophilus* to the reader.

Comment #6 of the reviewer3: Line 407 - Line 54 - Cognition is one of the functions of the brain. The authors should write in the Manuscript the idea that they study bacterial strain that has positive effects on brain function, which can be recognized through changes in cognitive processes.

Amendment for comment #6

Thanks for pointing out the most important part of understanding the experimental design (*Line 54-56,*

page 4 and *Line 411-415, page 21*). The content has been added to the Abstract and introduction so that the purpose of the study can be understood naturally.

Comment #7 of the reviewer3: Line 421 - why male?

Amendment for comment #7

We thank the reviewer for raising this issue. In an animal experiment, it is an ideal experiment by setting females and males as separate groups. However, our experiment was performed using only male subjects with consideration of some concerns.

- Simplification of the experimental variation affecting the interpretation of results¹
- Prevention of statistical power loss due to small subsamples for each sex²
- Estimation that there is no difference between the intestinal environment and the brain-gut axis system between female and male
- Male is mainly used in animal experiments for the brain-gut axis
- Restrictions on money, time, and the skill level of the experimenter.

Ideally, it is appropriate to use both males and females, but in consideration of these concerns, male mice were used. In order to provide this specific information to readers, this information was added to the manuscript (*Line 426-430, page 22*). We believe that it will help the reader's understanding.

Comment #8 of the reviewer3: Line 605 - Why is EG004 do not present here?

Amendment for comment #8

Thank you for your valuable advice. Since *L. acidophilus* EG004 was previously sequenced using the PacBio platform, only *Lcb. paracasei* EG005 and *Lcb. rhamnosus* EG006 were newly sequenced for this study. The sequence information of *L. acidophilus* EG004 is mentioned in *Line 634-636* and *page 31*, and related papers were cited to provide the sequencing information to readers. Also, since the paragraph indicated by the reviewer is about sequence information of the three strains, the sentence was changed to

‘Whole-genome sequencing of EG005 and EG006 and Whole-genome sequence of EG004’ (*Line 614, page 30*).

Comment #9 of the reviewer3: Line 623 - Add here the information from Data availability - The complete sequences of *Lcb. paracasei* EG005 and *Lcb. rhamnosus* EG006 is available in the NCBI database with accession numbers, SAMN23227569 and SAMN23227570, respectively. The metagenomic sequences are available in the NCBI database under the accession number PRJNA781018.

Amendment for comment #9

Thank you for the reviewer’s advice. Data availability information was added to the appropriate part (Whole-genome sequences of three probiotics; *Line 632-636, page 31*, and metagenomics data; *Line 586-587, page 30*).

Dept. of Agricultural Biotechnology and Research
Institute of Agriculture and Life Sciences
Seoul National University
San 56-1 Daehak-dong, Gwanak-gu, Seoul 151-742
Republic of Korea

References

- 1 Fields, R. D. J. N. NIH policy: mandate goes too far. *510*, 340-340 (2014).
 - 2 Richardson, S. S., Reiches, M., Shattuck-Heidorn, H., LaBonte, M. L. & Consoli, T. J. P. o. t. N. A. o. S. Opinion: focus on preclinical sex differences will not address women's and men's health disparities. *112*, 13419-13420 (2015).
-

December 7, 2021

Prof. Heebal Kim
Seoul National University
Seoul
Korea (South), Republic of

Re: Spectrum01815-21R2 (Positive effect on cognitive ability of *Lactobacillus acidophilus* EG004 in healthy mouse and fecal microbiome analysis using full-length 16S-23S rRNA metagenome sequencing)

Dear Prof. Heebal Kim:

Thanks for addressing the Reviewer's comments and congratulations on the acceptance of your manuscript for publication at Spectrum!

Your manuscript has been accepted, and I am forwarding it to the ASM Journals Department for publication. You will be notified when your proofs are ready to be viewed.

Sincerely,

Jan Claesen
Editor, Microbiology Spectrum

Journals Department
Supplemental material: Accept